# Enhancing Interpretability in Deep Reinforcement Learning through Semantic Clustering

**Liang Zhang**
College of Information Science
University of Arizona
Tucson, AZ 85721
liangzh@arizona.edu

**Justin Lieffers**
College of Information Science
University of Arizona
Tucson, AZ 85721
lieffers@arizona.edu

**Adarsh Pyarelal**
College of Information Science
University of Arizona
Tucson, AZ 85721
adarsh@arizona.edu

## Abstract

In this paper, we explore semantic clustering properties of deep reinforcement learning (DRL) to improve its interpretability and deepen our understanding of its internal semantic organization. In this context, semantic clustering refers to the ability of neural networks to cluster inputs based on their semantic similarity in the feature space. We propose a DRL architecture that incorporates a novel semantic clustering module that combines feature dimensionality reduction with online clustering. This module integrates seamlessly into the DRL training pipeline, addressing the instability of t-SNE and eliminating the need for extensive manual annotation inherent to prior semantic analysis methods. We experimentally validate the effectiveness of the proposed module and demonstrate its ability to reveal semantic clustering properties within DRL. Furthermore, we introduce new analytical methods based on these properties to provide insights into the hierarchical structure of policies and semantic organization within the feature space. Our code is available at https://github.com/ualiangzhang/semantic_rl.

## 1 Introduction

Deep reinforcement learning (DRL) has been widely applied in domains such as robotics, autonomous systems, game playing, and healthcare, due to its ability to solve complex decision-making tasks [1–3]. However, the black-box nature of DRL models obscures the decision-making process, potentially leading to unforeseen risks. In this study, we explore semantic clustering properties to improve the interpretability of DRL models. We use the term *semantic clustering* to refer to the process of grouping states eliciting similar agent behaviors under comparable environmental contexts (e.g., approaching a target, jumping to a higher platform in Procgen). Studying semantic clustering in DRL helps reveal the model's internal knowledge structure and semantic relationships between states, enhancing the interpretability and transparency of DRL models.

Although semantic clustering has been thoroughly investigated in natural language processing (NLP) [4, 5] and computer vision (CV) [6–9], it remains underexplored in DRL due to the complexity introduced by temporal dynamics and the absence of direct supervised signals. The sequential nature of decisions in DRL further complicates the task of capturing evolving semantics. Early work introduced *external* constraints—e.g., bisimulation [10, 11] and contrastive learning [12–15]—to

39th Conference on Neural Information Processing Systems (NeurIPS 2025).

shape feature spaces conducive to semantic clustering. In contrast, we focus on investigating whether DRL can *intrinsically* exhibit semantic clustering capabilities.

Mnih et al. [16] and Zahavy, Ben-Zrihem, and Mannor [17] analyzed the semantic distribution of the DRL feature space for Atari games using t-distributed Stochastic Neighbor Embedding (t-SNE) [18]. However, these studies are limited in multiple ways: (i) they are limited to a small set of Atari games with fixed scenes, making it difficult to distinguish whether clustering arises from pixel similarity or semantic understanding, (ii) Zahavy, Ben-Zrihem, and Mannor [17] manually define features for specific games, imposing substantial human effort, and (iii) both studies rely on t-SNE visualization for semantic analysis, which tends to produce unstable results and lacks an automated clustering mechanism. Thus, these approaches require significant manual effort for feature space annotation and analysis, hindering comprehensive semantic analysis and integration into downstream tasks.

Specifically, we make the following key contributions in this paper:

- We comprehensively explore the semantic clustering properties of DRL, advancing the understanding of the black-box decision-making processes. Unlike prior work that uses fixed-scene Atari games, we use Procgen[1] [19], which offers rich semantic diversity and dynamic, procedurally generated environments.
- We introduce a novel end-to-end architecture that integrates feature dimensionality reduction with online clustering, overcoming the limitations of prior t-SNE-based analyses and providing a more stable, effective means to study semantic properties in DRL.
- We present new analysis methods to reveal the internal semantic structure, uncover the hierarchical organization of policies, and identify potential risks in DRL models.

## 2 Related Work

**Semantic Clustering in NLP and CV** Prior work in NLP has shown that the spatial arrangement of word embeddings reflects semantic similarities, with semantically-related terms forming clusters in the embedding space [4, 5]. Similarly, in computer vision, images with similar content are positioned closely in the learned feature space [6–9].

**Semantic Clustering in DRL** Mnih et al. [16] and Zahavy, Ben-Zrihem, and Mannor [17] have previously explored visualizing the DRL feature space using t-SNE. In these studies, t-SNE visualizations show that features of states with close pixel distances tend to cluster together. However, due to the fixed nature of the scenarios they used (Atari games), semantic clustering could not be conclusively verified. This limitation motivates our use of Procgen to validate our approach.

**Interpretability of DRL** DRL interpretability research often focuses on video games due to their controlled environments and clear rules, which make analyzing decision-making processes easier. PW-Net [20] uses human-friendly prototypes to explain the model's decision-making. DIGR [21] generates saliency maps tlat highlight the most relevant features influencing the agent's decisions. Concept policy models integrate expert knowledge into multi-agent RL, enabling real-time intervention and interpretation of agent behavior [22]. MENS-DT-RL [23] applies decision trees to provide a rule-based explanation of the learning process. Furthermore, attention mechanisms and symbolic reasoning frameworks have also been applied to enhance interpretability [24–26]. Our work explores how DRL models internally organize information, offering new insights into the structure of learned representations.

**VQ-VAE** VQ-VAE [27] is a family of generative models that combine classic VAE with discrete latent representations through a posterior parameterization. Recently, VQ-VAE has been applied to various tasks, including high-resolution image generation [28], video generation [29], and speech coding [30]. It has also been used in model-based DRL to train transition models [31, 32].

## 3 Method

Our proposed architecture with a novel semantic clustering module is presented in Figure 1.

---

[1]Detailed environment instructions are available here, in Appendix A of Cobbe et al. [19], and in the repository.

**Background** The VQ-VAE workflow begins with an encoder network $\hat{E}$ that maps an input $\mathbf{x}$ to a latent representation $\hat{E}(\mathbf{x})$. This representation is then quantized by mapping it to the nearest embedding in a codebook $\{\mathbf{e}_k | k \in \{1, 2, \dots, K\}\}$. The quantized representation is then passed into a decoder network $\hat{D}$ to reconstruct the input $\mathbf{x}$. The loss function for VQ-VAE is defined as:

$$\mathcal{L}_{\text{VQ-VAE}} = \left\| \mathbf{x} - \hat{D}(\mathbf{e}_k) \right\|_2^2 + \left\| sg\left(\hat{E}(\mathbf{x})\right) - \mathbf{e}_k \right\|_2^2 + \beta \left\| sg(\mathbf{e}_k) - \hat{E}(\mathbf{x}) \right\|_2^2 \tag{1}$$

where $sg$ is a stop-gradient operator and $\beta$ weights the distance reduction between the encoded output $\hat{E}(\mathbf{x})$ and its closest embedding $\mathbf{e}_k$.

In this paper, we modify VQ-VAE to (i) assign features to the nearest VQ embedding for clustering, (ii) seamlessly integrate with DRL training, enabling simultaneous clustering and policy learning, and (iii) enhance clustering and interpretability through joint training with additional losses. Further details are provided in § 3.2.

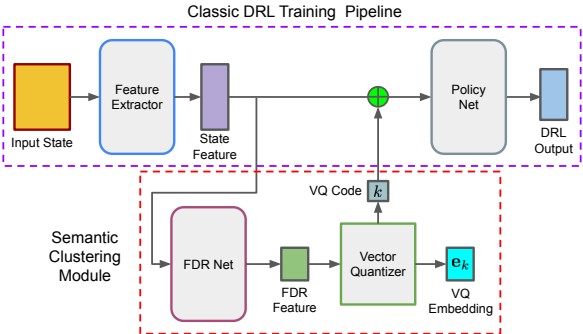

Figure 1: Overview of our architecture. The upper segment represents the classic DRL training pipeline, while the lower segment introduces the semantic clustering module. The Feature Dimensionality Reduction (FDR) net reduces the dimensionality of state features, resulting in FDR features, which the vector quantizer then processes to generate discrete VQ codes (denoted $k$)—which represent states associated with clusters—along with the closest VQ embeddings. Subsequently, $k$ is integrated into the state feature by element-wise addition after being expanded to match the state feature dimensions, enabling conditional policy training that better supports the integration of downstream tasks.

## 3.1 Semantic Clustering Module

To overcome the limitations of previous t-SNE-based semantic analyses (see § 1), we propose a novel semantic clustering module, which includes dimensionality reduction and online clustering.

**Dimensionality Reduction** Given the complexity of states in most DRL applications, their features are often high-dimensional. For example, Mnih et al. [16] use a 512-dimensional feature vector when training DQN on Atari games. Clustering high-dimensional features is challenging due to the curse of dimensionality [33, 34]. To mitigate these issues, we reduce feature dimensionality before clustering, resulting in more robust clustering outcomes. This not only simplifies the clustering process but also enables human-interpretable visualizations, typically in 2D.

The instability of t-SNE arises from its non-convex objective function, making it highly sensitive to initialization and leading to varied and unstable visualization outcomes [18, 35]. To overcome these challenges, we propose the *Feature Dimensionality Reduction* (FDR) network. This network remaps high-dimensional features to 2D using policy training data for online training, ensuring stable and efficient mappings after training. The FDR network's loss function is designed to preserve the consistency of *distance relationships* between high-dimensional and 2D feature spaces, measured by pairwise similarities as described in § 3.2.

**Online Clustering**

t-SNE–based analyses (e.g., [16, 17]) require per-state inspection and manual grouping because the plots lack clear cluster boundaries and often split semantically similar states across disconnected regions, making human curation time-consuming. To reduce such extensive annotation and facilitate

downstream integration, we introduce an online clustering approach—implemented via a modified VQ-VAE—that automatically segments the feature space and supports semantic analysis. This removes manual grouping: annotators instead watch a few short clips per discovered cluster and provide a semantic summary (see Table 1), typically within ≈15 minutes per environment. Details of the modified VQ-VAE design are provided in § 3.2.

## 3.2 Loss Function Design

The loss function for our proposed framework is given by

$$\mathcal{L}_{\text{total}} = \mathcal{L}_{\text{DRL}} + \lambda_{\text{ctrl}} \left( w_{\text{FDR}} \mathcal{L}_{\text{FDR}} + w_{\text{VQ-VAE}} \mathcal{L}'_{\text{VQ-VAE}} \right). \tag{2}$$

The DRL loss function $\mathcal{L}_{\text{DRL}}$ comes from the original DRL model. $w_{\text{FDR}}$ and $w_{\text{VQ-VAE}}$ are the weights of the FDR loss ($\mathcal{L}_{\text{FDR}}$) and the modified VQ-VAE loss ($\mathcal{L}'_{\text{VQ-VAE}}$), respectively. $\lambda_{\text{ctrl}}$ represents the control factor. We explain each of these components below.

**FDR Loss** $\mathcal{L}_{\text{FDR}}$ is based on state features from the DRL training batch and FDR features generated by the FDR network. We use the Student's $t$-distribution for pairwise similarities as it captures nonlinear structures and efficiently measures pairwise relative positions of features within a batch without requiring the entire feature set, making it ideal for online clustering.

The pairwise similarities of state features $p_{ij}$ are given by

$$p_{ij} = \frac{d(i, j)}{\sum_{k \neq l} d(k, l)}, \quad \text{where } d(m, n) = \left( 1 + \frac{\|f(\mathbf{s}_m) - f(\mathbf{s}_n)\|^2}{\alpha} \right)^{-\frac{\alpha+1}{2}}. \tag{3}$$

Here, $f$ is the feature extractor, $\mathbf{s}_i$ is the $i^{\text{th}}$ state in a batch, $\alpha$ is the Student's-$t$ degrees-of-freedom parameter, and $\| \cdot \|$ is the $\ell_2$ norm. The pairwise similarities for FDR features, $q_{ij}$ are computed using the same expression as (3), but with $f$ replaced by $g \circ f$, where $g$ is the FDR net. In contrast to other deep clustering studies, e.g., Xie, Girshick, and Farhadi [36] and Li, Qiao, and Zhang [37], the same degree of freedom $\alpha$ is selected for both high- and low-dimensional similarities, ensuring that the original distance relationship between features is maintained in the low-dimensional space.

The FDR loss is given by

$$\mathcal{L}_{\text{FDR}} = - \sum_i \sum_j p_{ij} \log \left( q_{ij} \right). \tag{4}$$

Minimizing $\mathcal{L}_{\text{FDR}}$ encourages the low-dimensional mapping to preserve the pairwise neighborhood structure of the high-dimensional features.

**Modified VQ-VAE Loss** To perform clustering, we use the second term of $\mathcal{L}_{\text{VQ-VAE}}$ from (1), which moves VQ embeddings closer to neighboring FDR features. These embeddings function similarly to centroids in online $k$-means [38] clustering. Since the other terms are unnecessary for our model, we only retain and modify the second term to define the modified VQ-VAE loss:

$$\mathcal{L}'_{\text{VQ-VAE}} = \|sg \left[ g \left( f \left( \mathbf{s} \right) \right) \right] - \mathbf{e}_k\|_2^2, \tag{5}$$

where $\mathbf{e}_k$ is the closest embedding in the codebook to the FDR feature $g(f(\mathbf{s}))$.

**Control Factor** Since effective semantic clustering relies on a clear and distinguishable semantic distribution that is often difficult to achieve in the early stages of training, we propose an adaptive control factor ($\lambda_{\text{ctrl}}$) strategy updated according to training performance (see Appendix A).

**Improved Clustering** Our loss design not only achieves dimensionality reduction and clustering but also enhances clustering properties, making the states within each cluster more compact (smaller intra-cluster distances) and the cluster boundaries more separable. This is crucial for clearly distinguishing the semantics of states at the cluster boundaries, further enhancing the model's interpretability. Because of the stop-gradient in $\mathcal{L}'_{\text{VQ-VAE}}$, it does not directly pull FDR features toward their nearest codebook embeddings. However, when the FDR features become denser during joint training with $\mathcal{L}_{\text{FDR}}$, $\mathcal{L}'_{\text{VQ-VAE}}$—and thus $\mathcal{L}_{\text{total}}$—decreases, yielding tighter clusters. Moreover, since

$\mathcal{L}_{\text{FDR}}$ aligns the affinity matrices $p$ and $q$, this densification in the low-dimensional FDR space is reflected and propagates into the high-dimensional state features. We demonstrate the improved clustering in § 4 and provide more evidence of this enhanced clustering property and the intrinsic nature of semantic clustering in DRL in § C.2.

---

**Algorithm 1:** PPO with Semantic Clustering Module (SCM)

**Input:** PPO network parameters $\theta$, FDR network parameters $\phi$, SCM hyperparameters, and PPO hyperparameters such as value loss weight $w_{\text{value}}$, entropy loss weight $w_{\text{entropy}}$.

1 **for** *each training iteration $i = 1, 2, \ldots$* **do**
2     Collect $N$ trajectories $\mathcal{D}_i = \{\tau_1, \ldots, \tau_N\}$ using policy $\pi_\theta$;    // Trajectory collection
3     **for** *each epoch $j = 1, 2, \ldots$* **do**
4         **for** *each minibatch $M \subseteq \mathcal{D}_i$* **do**
5             **for** *each state $\mathbf{s}_m \in M$* **do**
6                 $\mathbf{f}_m \leftarrow f_\theta(\mathbf{s}_m)$;     // Extract state feature
7                 $\mathbf{f}_m^{\text{FDR}} \leftarrow g_\phi(\mathbf{f}_m)$;     // Extract FDR feature
8                 $k_m \leftarrow \arg\min_k \|\mathbf{f}_m^{\text{FDR}} - \mathbf{e}_k\|$;     // Assign to nearest VQ embedding
9                 $\mathbf{k}_m^{\text{expand}} \leftarrow \text{expand}(k_m, \dim(\mathbf{f}_m))$;     // Broadcast to state feature dim.
10                 $\mathbf{f}_m^{\text{fused}} \leftarrow \mathbf{f}_m + \mathbf{k}_m^{\text{expand}}$;     // Apply element-wise addition
11                 $\pi(a|\mathbf{s}_m) \leftarrow \hat{\pi}_\theta(\mathbf{f}_m^{\text{fused}})$;     // Compute policy outputs
12                 $V(\mathbf{s}_m) \leftarrow \hat{V}_\theta(\mathbf{f}_m^{\text{fused}})$;     // Compute value outputs
13             $\mathcal{L}_{\text{PPO}} \leftarrow \mathcal{L}_{\text{policy}} + w_{\text{value}}\mathcal{L}_{\text{value}} + w_{\text{entropy}}\mathcal{L}_{\text{entropy}}$;     // PPO loss
14             $\mathcal{L}_{\text{SCM}} \leftarrow w_{\text{FDR}}\mathcal{L}_{\text{FDR}} + w_{\text{VQ-VAE}}\mathcal{L}'_{\text{VQ-VAE}}$;     // SCM loss
15          $\mathcal{L}_{\text{total}} \leftarrow \mathcal{L}_{\text{PPO}} + \lambda_{\text{ctrl}}\mathcal{L}_{\text{SCM}}$;     // Total loss
16         Update $\theta$, $\phi$, and $\{\mathbf{e}_k\}_{k=1}^K$ by minimizing $\mathcal{L}_{\text{total}}$;     // Parameter update

---

**Advantages of Online Training** Online training offers several advantages: (i) it enhances clustering by incorporating the training of $\mathcal{L}_{\text{total}}$, (ii) training the VQ code $k$ with a latent-conditioned policy $\pi(a|\mathbf{s}, k)$ (where $a$ is the action) supports extension to downstream tasks, such as macro action selection in hierarchical learning, and (iii) it improves memory efficiency by eliminating the need to store a large number of states during model training.

**Training Process** The training process of our framework builds upon the structure of the original DRL algorithm while incorporating the semantic clustering module (SCM) by using (2) for total loss calculation. We take PPO [39] as an example, and the training procedure is outlined in algorithm 1.

## 4 Simulations

In this work, we primarily study the intrinsic characteristics and black-box decision-making of DRL, and address the instability of t-SNE visualizations used in prior studies. Therefore, this section aims to: (i) compare t-SNE to validate the stability and effectiveness of the proposed clustering method, (ii) assess the semantic clustering properties of DRL to improve interpretability, and (iii) introduce new methods to analyze policies and internal model characteristics, identifying issues in DRL decision-making. The integration of our module has minimal impact on performance (see § C.1 and Appendix D).

### 4.1 Clustering Effectiveness Evaluation

We demonstrate the clustering effectiveness of our proposed approach using the CoinRun game from Procgen as an example. Similar results can be easily extended to other games using the code and checkpoints provided in the supplementary material. We use a trained model to collect states, where the agent selects actions randomly with a probability of 0.2 and follows the trained policy with a probability of 0.8 to ensure diverse state coverage. States are sampled with a probability of 0.8, and 64 parallel environments collect states over 500 steps, resulting in ≈25,000 states for visualization.

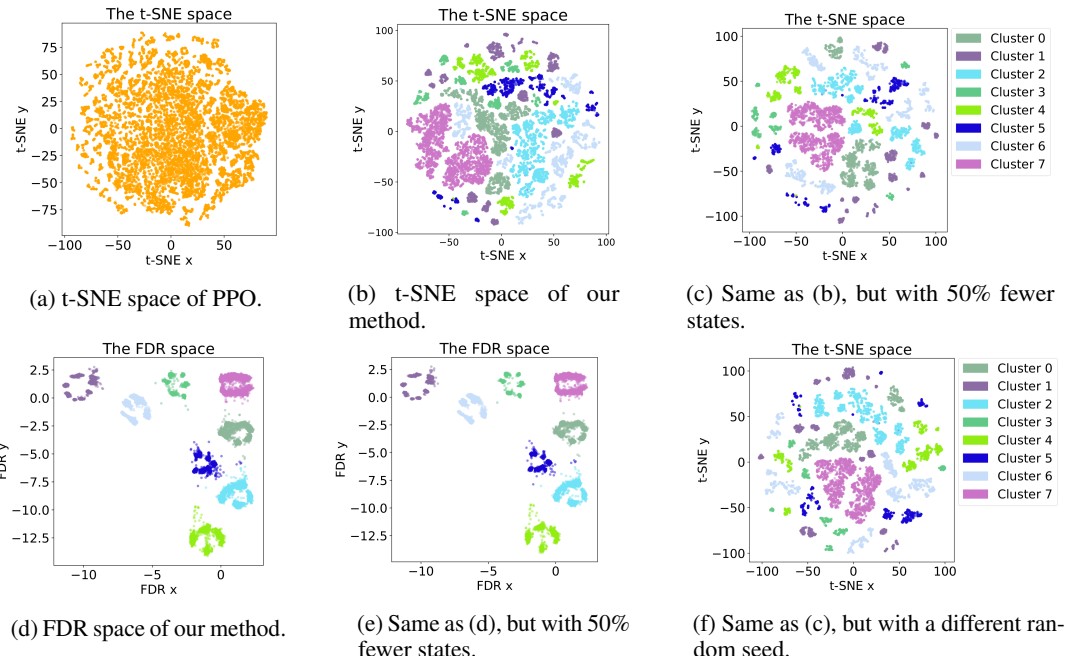

(a) t-SNE space of PPO.

(b) t-SNE space of our method.

(c) Same as (b), but with 50% fewer states.

(d) FDR space of our method.

(e) Same as (d), but with 50% fewer states.

(f) Same as (c), but with a different random seed.

Figure 2: Visualization of features in t-SNE and FDR spaces using PPO and our method. To enable comparison, feature colors in the t-SNE visualizations of our method correspond to the cluster colors in the FDR space, while PPO features are shown in orange due to the absence of clustering. Unlike t-SNE, which fails to produce clearly separable clusters and exhibits sensitivity to the number of states and random seeds, our method yields well-separated and stable clusters under varying conditions.

Note that the cluster colors (indices) in the t-SNE plots are assigned by our method and are used solely to facilitate comparison of spatial relationships.

**Cluster Separation and Improved Clustering** The t-SNE visualization of PPO (Figure 2a), spreads features across the space without forming clear clusters, limiting its utility for clustering analysis and requiring detailed manual examination of certain areas, as in previous studies. In contrast, the t-SNE visualization of our method (Figure 2b), reveals numerous distinct, small clusters. States within each of these clusters originate from the same semantic group identified by our method. This dispersion into multiple smaller clusters is due to t-SNE's focus on local structures and its tendency to avoid crowding, causing complete semantic clusters to scatter. The visualization in the FDR space (Figure 2d), displays clear and separate complete clusters, which are identified by VQ codes. § C.2 presents a stop-gradient ablation that disables our proposed module while keeping the backbone and training protocol fixed; visualizations show fuzzier, less separable clusters than the full model, further supporting our gains in sharpness and coherence.

**Sensitivity to Number of States** Our method's stability is showcased in Figures 2c and 2e, where the number of processed states is reduced by 50%. Unlike the drastic changes in feature distribution seen in the t-SNE space (Figure 2c), the FDR space (Figure 2e) exhibits a stable mapping, merely reducing the quantity of features without altering their spatial distribution.

**Sensitivity to Random Seed** While the t-SNE representation is sensitive to randomness, as demonstrated by the significant difference between Figures 2c and 2f, the FDR space's mapping remains unchanged even when the random seed is altered, maintaining the distribution in Figure 2e. t-SNE's randomness primarily stems from its random initialization and non-convex optimization process, leading to significantly different visualizations with different random seeds. In contrast, our model produces a stable feature mapping after training, which does not vary with random seeds.

These clear contrasts highlight the robustness of FDR over the instability of t-SNE, addressing prior limitations and enabling stable semantic clustering and analysis. In Appendix E, we further present a statistical comparison of common dimensionality reduction methods across multiple clustering metrics, demonstrating that our method achieves superior clustering performance.

## 4.2 Semantic Clustering in DRL

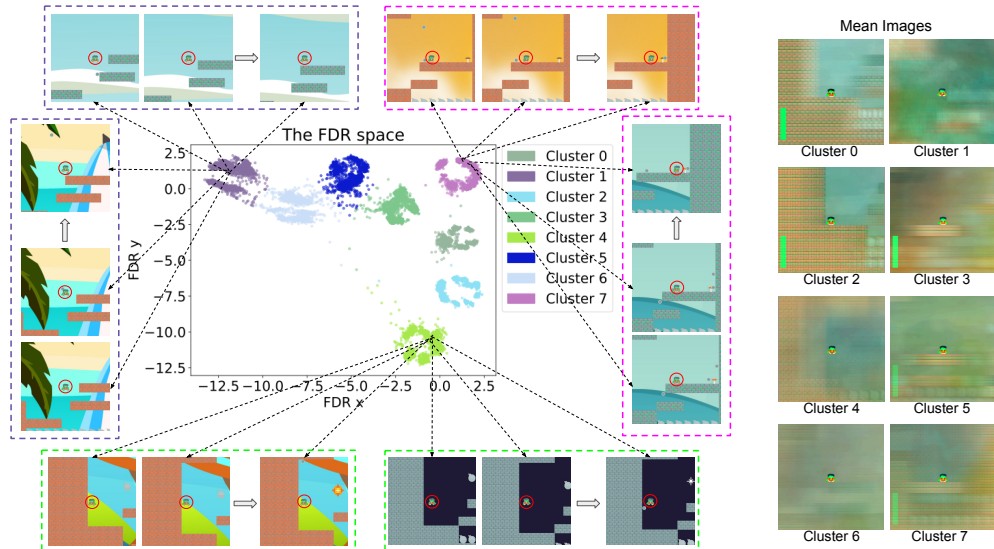

Figure 3: State examples in the Ninja FDR space and the mean images of clusters. Each dashed box contains a sequence of consecutive states assigned to the same cluster, with dotted arrows indicating their corresponding FDR feature positions. These examples demonstrate that semantically similar and temporally adjacent states are grouped into the same cluster, highlighting the learned semantic coherence. Descriptions of the state sequences in the clusters are provided in Table 1.

Table 1: Cluster descriptions and mean image outlines for the Ninja game

| Cluster | Description | Mean image outlines |
|---------|-------------|---------------------|
| 0 | The agent starts by walking through the first platform and then performs a high jump to reach a higher ledge. | Essential elements are outlined, e.g., a left-side wall, the current position of the agent on the first platform, and the upcoming higher ledges. |
| 1 | The agent makes small jumps in the middle of the scene. | We can observe the outlines of several ledges below the agent. |
| 2 | Two interpretations are present: 1) the agent starts from the leftmost end of the scene and walks to the starting position of Cluster 0, and 2) when there are no higher ledges to jump to, the agent begins from the scene, walks over the first platform, and prepares to jump to the subsequent ledge. | The scene prominently displays the distinct outline of the left wall and the first platform. The agent's current position is close to both of them. |
| 3 | The agent walks on the ledge and prepares to jump to a higher ledge. | The agent is standing on the outline of the current ledge and the following higher ledges. |
| 4 | After performing a high jump, the agent loses sight of the ledge below. | The agent is performing a high jump. |
| 5 | The agent walks on the ledge and prepares to jump onto a ledge at the same height or lower. | The agent is standing on the outline of the current ledge and the following ledges at the same height or lower. |
| 6 | The agent executes a high jump while keeping the ledge below in sight. | The agent is performing a high jump and the outline of the ledge below is visible. |
| 7 | The agent moves towards the right edge of the scene and touches the mushroom. | The outlines of the wall and platform on the far right are visible. |

In this section, we illustrate semantic clustering analysis using the Ninja game, in which the agent goes from left to right, jumping over various ledges and scores points by touching the mushroom on the far right. In Appendix G, we analyze additional games, reaching similar conclusions.

**Mean Image Analysis**  We performed a qualitative analysis of the mean images of states within each cluster. Figure 3 presents state examples from the FDR space of Ninja along with the mean images of each semantic cluster, and Table 1 contains natural language descriptions of the clusters as well as notable features of the mean images corresponding to each cluster. Corresponding videos can be found in the supplementary material.

Unlike *static* semantic clustering in some CV and NLP tasks, where clustering is based on a single image or word, DRL's semantic clustering is *dynamic* in nature—state sequences with similar semantics are grouped into the same semantic cluster. Notably, this semantic clustering goes beyond pixel distances and operates on a *semantic understanding* level of the environment, as illustrated in figures 3 and 4. This generalized semantic clustering emerges from the DRL model's inherent ability to learn and summarize from changing scene dynamics, independent of external constraints like bisimulation or contrastive learning, and without the need for supervised signals. The neural network's internal organization of policy-relevant knowledge indicates clustering-based spatial organization based on semantic similarity. Furthermore, we find that video sequences within clusters can be summarized using natural language, akin to the 'skills' humans abstract during learning processes.

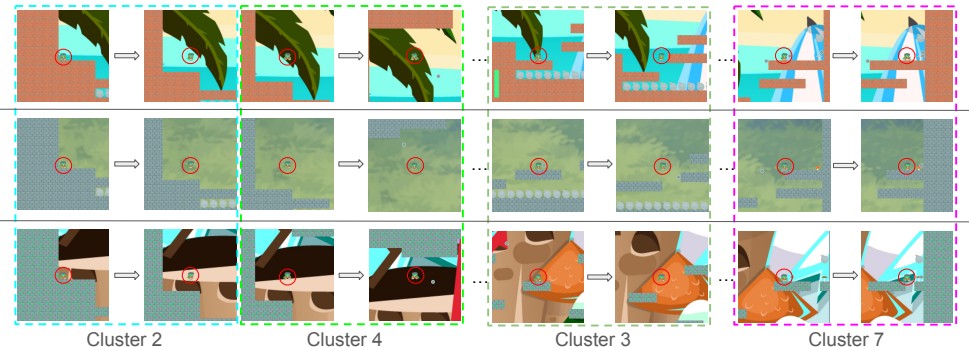

Figure 4: Three episodes from the Ninja game. States within colored dashed boxes correspond to clusters of the same colors in Figure 3. Solid gray arrows indicate omitted intermediate states from the same cluster, while ellipses represent other omitted states. These visualizations illustrate consistent semantic alignment in cluster assignments across different episodes.

Table 2: Human evaluation results

| No. | Statement | Mean Score (SEM) | | |
| --- | --- | --- | --- | --- |
| | | Jumper | FruitBot | Ninja |
| 1 | *The clips of each cluster consistently display the same skill being performed* | 4.24 (0.15) | 4.10 (0.11) | 4.30 (0.15) |
| 2 | *The clips of each cluster match the given skill description* | 4.36 (0.16) | 4.16 (0.11) | 4.20 (0.17) |
| 3 | *The identified skills aid in understanding the environment and the AI's decision-making process* | 4.50 (0.22) | 4.10 (0.18) | 4.20 (0.20) |

**Human Evaluation**  In addition to qualitatively analyzing the mean images, we hired 15 human evaluators to validate the semantic clustering properties. Evaluators were adults (18+), native or highly proficient English speakers, with basic video game experience and a brief training session. Specifically, video sequences from each episode are segmented into multiple clips based on the cluster each frame belongs to, and these clips are grouped by cluster for evaluators to review. Each evaluator watched these grouped clips and responded to three interpretability-related statements for two out of a set of three games (Jumper, Fruitbot, and Ninja). The response for each question was chosen from a five-point Likert scale with the options: *Strongly Disagree (1)*, *Disagree (2)*, *Neutral (3)*, *Agree (4)*, and *Strongly Agree (5)*. Further details on the evaluation procedure are provided in Appendix I.

The statements and the results of the human evaluation are provided in Table 2. The mean scores for all statement-environment combinations are greater than 4, with the exception of statements 1 and 3 for FruitBot, for which the lower bounds on the mean set by the standard error of the mean (SEM) are 3.99 and 3.92 respectively. The slightly lower score on FruitBot may be caused by the behavior description of clusters, the agent's relative distance to the wall ahead (far/near) and the agent's relative position on the screen (left/center/right) require a higher degree of subjective judgment. In contrast, Jumper has a clear radar for direction and position information, and Ninja has more explicit behavioral reference objects, e.g., ledges and mushrooms. Overall, these results suggest that humans generally agree that our model possesses semantic clustering properties and supports interpretability.

## 4.3 Model and Policy Analysis

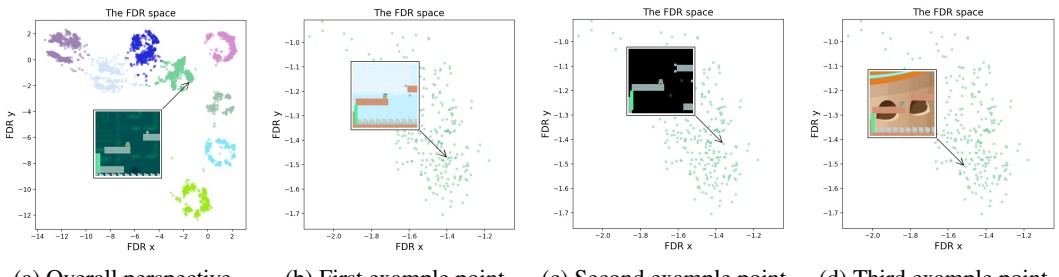

(a) Overall perspective.  (b) First example point  (c) Second example point  (d) Third example point

Figure 5: Hover examples in the FDR space of Ninja. We observe a sub-cluster in the FDR space as an example from an zoomed-out perspective (a) and zoomed-in perspectives (b), (c), and (d). The agent is standing on the edge of a ledge. Although the scenarios of (b), (c), and (d) are different, the proposed method effectively clusters semantically consistent features together in the FDR space.

To better explore the knowledge organization within the internal space of DRL models, we developed a visualization tool (see Figure 5 for an example). The tool supports 'statically' analyzing the semantic distribution of models—specifically, (i) when the mouse cursor hovers over a specific feature point, the corresponding state image is displayed, and (ii) the tool includes a zooming functionality to observe the semantic distribution of features in detail within clusters.

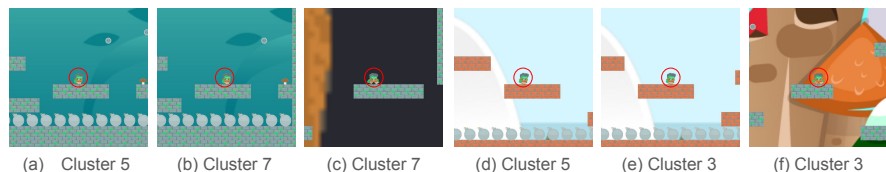

(a)  Cluster 5    (b) Cluster 7    (c) Cluster 7    (d) Cluster 5    (e) Cluster 3    (f) Cluster 3

Figure 6: Policy analysis examples in Ninja, showing states assigned to different clusters.

In addition, we propose a more 'dynamic' analysis method—the VQ code enables us to determine the cluster to which the current state belongs, which allows for the semantic segmentation of episodes, as exemplified in Figure 4. Our model excels at breaking down complex policies, thereby shedding light on their inherent hierarchical structures. Moreover, this segmentation is based on semantics, making it understandable to humans and likely to improve interpretability in downstream hierarchical learning tasks. Consequently, this method introduces a 'dynamic' strategy for dissecting policy structures.

We present policy analysis examples in Figure 6, leveraging clustering results from our method. Figures 6(a) and 6(b) show consecutive states assigned to clusters 5 and 7, respectively (see Figure 3 and Table 1). In 6(a), the right-side wall is absent and the agent walks along the ledge (cluster 5); in 6(b), a right-side wall appears and the agent transitions to cluster 7, approaching the mushroom. Because the mushroom is visible in both frames, this cluster change is driven by detecting the right-side wall rather than by the mushroom's presence. This finding is confirmed in 6(c), which shows another state from cluster 7 without any mushroom present, where the agent continues along the ledge, incorrectly perceiving the conditions for cluster-7 behavior. Figures 6(d) and 6(e) depict states within the same episode: in 6(d), the agent initially plans to jump onto a lower ledge (cluster 5), but upon seeing a higher, safer ledge in 6(e), it shifts its strategy accordingly (cluster 3). Similarly,

6(f) shows a state that has been assigned to cluster 3 by our model, helping us anticipate the agent's future behavior of jumping onto the higher ledge. These analyses demonstrate how our model helps clarify policy behaviors, uncover decision-making structures, and identify potential issues.

# 5    Limitations and Future Work

Our approach has several limitations to be addressed in future work. First, it relies on clear semantic distributions, which can become unstable when policies deviate significantly from optimal behavior, resulting in ambiguous clusters. More robust clustering methods may be needed to improve stability. Second, as the method is unsupervised, selecting an appropriate number of clusters is crucial— too few clusters reduces clarity, while too many clusters causes semantic fragmentation. We used eight clusters to balance interpretability and granularity, but future work could explore adaptive techniques that adjust the number of clusters based on task complexity, the elbow method, silhouette-score optimization, etc. For further analysis of the impact of cluster numbers on performance and interpretability, see Appendix D. Furthermore, policy interpretations are manually described. In future work, we aim to automate behavior summarization and explanation (e.g., using GPT-4V). In addition, since the FDR module optimizes pairwise similarities, a natural extension is to replace the current affinity with alternative measures (e.g., cosine similarity or bisimulation metrics). Lastly, we plan to extend this method to other DRL algorithms, benchmarks, and settings.

# 6    Conclusion

In this paper, we investigated the semantic clustering properties of DRL. Using a novel approach that combines dimensionality reduction and online clustering, we analyzed the internal organization of knowledge within the feature space. Our method provides a stable mapping of feature positions and enhances semantic clustering, revealing meaningful structures in continuous sequences of video game states. We demonstrate that semantic clustering in DRL arises dynamically as the agent interacts with its environment. As the agent explores diverse states during reinforcement learning, it naturally clusters semantically related states based on spatial and temporal relationships. This dynamic clustering exploits regularities in the environment, offering a unique approach compared to the static clustering observations in NLP and CV.

## Acknowledgments

Research was sponsored by the Army Research Office and was accomplished under awards W911NF-20-1-0002 and W911NF-24-2-0034. The views and conclusions contained in this document are those of the authors and should not be interpreted as representing the official policies, either expressed or implied, of the Army Research Office or the U.S. Government. The U.S. Government is authorized to reproduce and distribute reprints for Government purposes notwithstanding any copyright notation herein. We gratefully acknowledge the anonymous reviewers and the program committee for their thoughtful feedback and constructive suggestions, which helped improve this work. We also thank Huy Le and Robert Lopez for their valuable assistance with the human evaluation.

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

## Appendix Overview

## A   Architecture, Hyperparameters, and Computational Costs

The training of the proposed method is consistent with the Impala architecture [40], the PPO algorithm [39], and the hyperparameters used in the Procgen paper [19]. The FDR net is composed of two fully connected layers with 128 and 2 neurons, respectively. The codebook in the vector quantizer has eight embeddings, and the degree of freedom in the FDR loss is 20. The random seeds employed in Figure 2 are 2021 and 2031, while the seeds used in Figure C.1 are 2021, 2022, and 2023. We train all models on one NVIDIA Tesla V100S 32GB GPU. The operating system version is CentOS Linux release 7.9.2009. Each runs takes around six hours.

In Equation 2 of the main paper, $w_{\text{FDR}}$ and $w_{\text{VQ-VAE}}$ are 500 and 1, respectively. $\lambda_{\text{ctrl}}$ is updated every 50 iterations according to the following expression:

$$\lambda_{\text{ctrl}} = \min\left(\frac{s_{\text{mean}}}{0.8 \cdot s_{\text{highest}}}, 1\right), \tag{A.1}$$

where $s_{\text{mean}}$ is the mean score of the last 100 episodes in training, and $s_{\text{highest}}$ is the highest score of the environment.

All hyperparameters introduced in our method, except for the number of embeddings, were chosen through performance tuning to optimize the model's overall performance. The number of embeddings in the vector quantizer was determined by ensuring that each cluster maintained a singular semantic interpretation. During hyperparameter tuning, we found that performance is primarily influenced by $w_{\text{FDR}}$, $w_{\text{VQ-VAE}}$, and $\lambda_{\text{ctrl}}$, and is more robust to the number of VQ embeddings and the degrees of freedom in Equation 3.

## B   Theoretical Analysis of Loss Design

The two auxiliary losses in our framework serve distinct theoretical purposes. The $\mathcal{L}_{\text{FDR}}$ term is intended to preserve relative geometry when mapping high-dimensional state features to a 2-D space, whereas the modified VQ term should behave like a standard online $k$-means step so that the codebook converges to meaningful cluster centroids. We formalize both claims below.

### B.1   Batch-wise Distance Similarity Preservation

**Goal.**   We first prove that driving $\mathcal{L}_{\text{FDR}} \to 0$ guarantees that pairwise similarity orderings are preserved between the original feature space and the FDR space, thereby retaining semantic neighborhood structure.

**Notation.**   Let

- $n$ be the mini-batch size.
- $\{x_i\}_{i=1}^n$ be the batch of high-dimensional state features.
- $\{y_i\}_{i=1}^n$ be the corresponding low-dimensional embeddings produced by the FDR network.

- $\alpha > 0$ be the degrees of freedom of the Student–$t$ kernel.
- The Student–$t$ kernel

$$d_{\mathrm{t}}(u, v) = \left(1 + \frac{\|u-v\|^2}{\alpha}\right)^{-\frac{\alpha+1}{2}}, \quad d_{\mathrm{t}}(u, v) \in (0, 1].$$

Define for all $1 \leq i \neq j \leq n$ the normalized pairwise similarities

$$p_{ij} = \frac{d_{\mathrm{t}}(x_i, x_j)}{\sum_{k \neq \ell} d_{\mathrm{t}}(x_k, x_\ell)}, \qquad q_{ij} = \frac{d_{\mathrm{t}}(y_i, y_j)}{\sum_{k \neq \ell} d_{\mathrm{t}}(y_k, y_\ell)},$$

and the Kullback–Leibler divergence

$$\mathcal{L}_{\mathrm{FDR}} = \sum_{i \neq j} p_{ij} \log \frac{p_{ij}}{q_{ij}} \geq 0.$$

**Theorem 1** (Similarity Preservation). *If $\mathcal{L}_{\mathrm{FDR}} = 0$, then there exists a constant*

$$\kappa = \frac{\sum_{k \neq \ell} d_{\mathrm{t}}(y_k, y_\ell)}{\sum_{k \neq \ell} d_{\mathrm{t}}(x_k, x_\ell)} > 0$$

*such that for all $i \neq j$,*

$$d_{\mathrm{t}}(y_i, y_j) = \kappa \, d_{\mathrm{t}}(x_i, x_j).$$

*Consequently, the ordering of squared distances $\|y_i - y_j\|^2$ and $\|x_i - x_j\|^2$ is identical. Moreover, if $\kappa = 1$, then $d_{\mathrm{t}}(y_i, y_j) = d_{\mathrm{t}}(x_i, x_j)$ and hence $\|y_i - y_j\|^2 = \|x_i - x_j\|^2$.*

*Proof.* By the non-negativity of KL divergence, $\mathcal{L}_{\mathrm{FDR}} = 0$ iff $p_{ij} = q_{ij}$ for every $i \neq j$. Equating

$$\frac{d_{\mathrm{t}}(y_i, y_j)}{\sum_{k \neq \ell} d_{\mathrm{t}}(y_k, y_\ell)} = \frac{d_{\mathrm{t}}(x_i, x_j)}{\sum_{k \neq \ell} d_{\mathrm{t}}(x_k, x_\ell)} \implies d_{\mathrm{t}}(y_i, y_j) = \kappa \, d_{\mathrm{t}}(x_i, x_j).$$

Since the Student-$t$ kernel decreases strictly with $\|u - v\|^2$, scaling by $\kappa > 0$ preserves rank order. When $\kappa = 1$, strict monotonicity forces equality of squared distances. $\square$

Minimizing $\mathcal{L}_{\mathrm{FDR}}$ therefore enforces a batch-wise isometry up to a global scale factor $\kappa$, which is exactly the property needed for semantic clustering in the 2-D FDR space.

### B.2 Modified VQ–$k$-Means Equivalence

**Goal.** Next we show that the gradient update used for the vector-quantizer codebook is algebraically identical to an online $k$-means step.

**Notation.** Let

- $y_i = g(f(x_i))$ be the FDR feature for state $x_i$.
- $\{e_k\}_{k=1}^K$ be the learnable codebook entries.
- $k_i = \arg\min_{1 \leq k \leq K} \|y_i - e_k\|_2$ be the nearest code index.
- $m_k = |\{i : k_i = k\}|$ be the cluster size.
- $\bar{y}_k = \frac{1}{m_k} \sum_{i:k_i=k} y_i$ be the empirical cluster centroid.
- The modified VQ loss $\mathcal{L}'_{\mathrm{VQ}} = \sum_{i=1}^n \|\mathrm{sg}(y_i) - e_{k_i}\|_2^2$.

**Theorem 2** (Equivalence to Online $k$-Means). *Updating each codebook vector via*

$$e_k^{(t+1)} = e_k^{(t)} - \beta \, \nabla_{e_k} \mathcal{L}'_{\mathrm{VQ}}, \qquad \beta > 0,$$

*produces*

$$e_k^{(t+1)} = e_k^{(t)} + \gamma_k (\bar{y}_k - e_k^{(t)}), \qquad \gamma_k = 2\beta \, m_k,$$

*which is the online k-means update with learning rate $\gamma_k$.*

*Proof.* Because stop-gradient sg($y_i$) blocks gradients with respect to $y_i$,

$$\nabla_{e_k} \mathcal{L}'_{\mathrm{VQ}} = 2 \sum_{i:k_i=k} (e_k - y_i).$$

Therefore

$$e_k^{(t+1)} = e_k^{(t)} - 2\beta \sum_{i:k_i=k} \left(e_k^{(t)} - y_i\right) = e_k^{(t)} + 2\beta\, m_k\left(\bar{y}_k - e_k^{(t)}\right),$$

which matches the standard incremental $k$-means rule. $\qquad\square$

## C  Ablation Study on Performance and Interpretability

### C.1  Performance Impact of the Semantic Clustering Module

Considering the cost of time and computational resources, we opt for training our model on the full distribution of levels in the 'easy' mode. In Figure C.1, a comparison of performance curves between the proposed method and the baseline is presented, where 'SPPO' denotes 'semantic' PPO, i.e., PPO integrated with our proposed semantic clustering module. Consistent with the Procgen paper [19], given the diversity of episodes during training, a single curve represents both training and testing performance. Across these environments we observe that the proposed method closely aligns with the baseline performance, indicating minimal impact on performance from the introduced module. This is expected, as the module only performs dimensionality reduction and clustering based on existing features, without introducing external information. The discrete code $k$ reflects only the position of a feature in the learned space and is expanded and added element-wise to preserve the original feature dimensionality, ensuring the policy receives no additional information beyond what is already contained in the state feature.

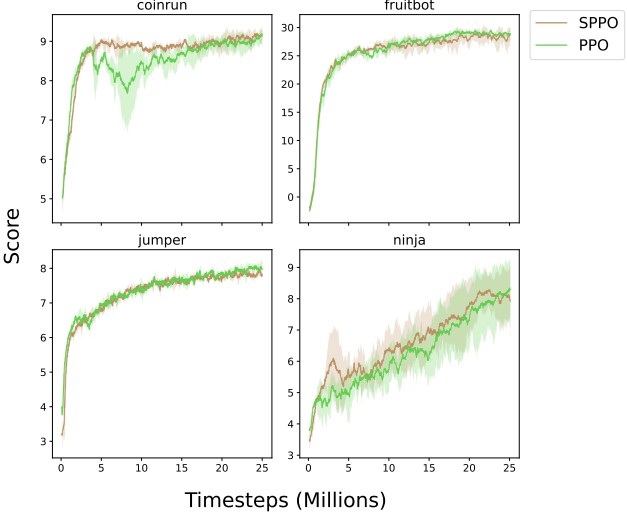

Figure C.1: Performance curves on 'easy' difficulty environments using three random seeds, trained and evaluated on the full distribution of levels.

### C.2  Semantic Clustering as an Intrinsic Property of DRL

We conducted a stop gradient experiment to further investigate whether semantic clustering is an inherent property of DRL. In this experiment, we applied a stop gradient operation to Equation 4 and removed the connection between the VQ codes and the original state features. This was done to prevent the semantic clustering module from influencing the feature space and to observe whether semantic clustering would still occur. The results, as shown in Figures C.2 and C.3, demonstrated that states within the same semantic cluster continued to exhibit similar semantic interpretations, even without the influence of the semantic clustering module. However, the boundaries between clusters became less clear, making it more difficult to distinguish the semantics of states near the edges

of clusters. Notably, we also observed that this modification had minimal impact on performance, consistent with the trend shown in Figure C.1.

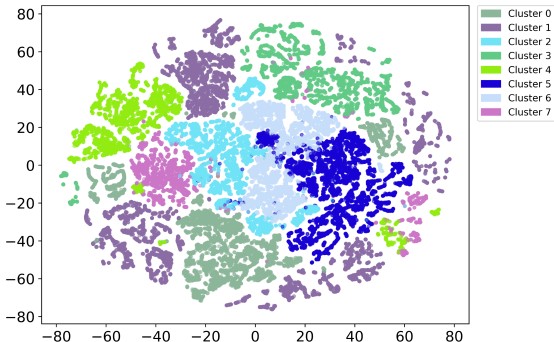

Figure C.2: Visualization of Features in the t-SNE Space. The training eliminates the impact of the proposed semantic clustering module on the original feature space. Feature colors correspond to cluster colors in the FDR space of Figure C.3, facilitating the comparison of spatial relationships and feature distribution changes. Compared to Figure 2b in the main paper, the absence of the semantic clustering module's enhancement makes sub-clusters less distinct.

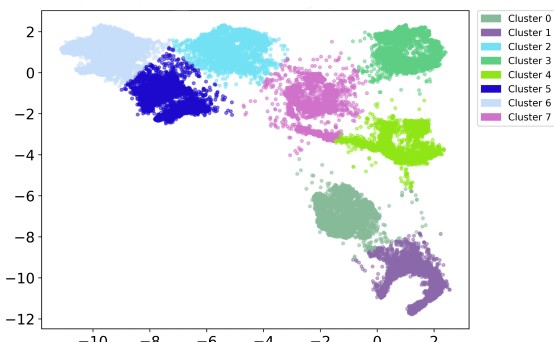

Figure C.3: Visualization of Features in the FDR Space. The training eliminates the impact of the proposed semantic clustering module on the original feature space. Compared to Figure 2d in the main paper, the cluster boundaries in the FDR space are less distinct.

These observations suggest that semantic clustering is indeed an intrinsic property of DRL, driven by the agent's interaction with its environment during training. The proposed semantic clustering module enhances this natural clustering behavior by increasing the density of clusters, thus improving the separability between them. To fine-tune the influence of the module, we introduced a control factor. At the beginning of training, the control factor is kept low, allowing the DRL training to shape the feature space independently. As the policy becomes more optimized and the semantic distribution of states becomes more organized, the control factor is gradually increased to further enhance the clarity and separability of clusters.

## D   Impact of the Number of VQ Embeddings on Performance and Interpretability

To analyze the effect of the number of VQ embeddings ($K$) on both model performance and interpretability, we conducted experiments using the *Jumper* environment as an example. Similar conclusions can be extended to other environments.

## D.1  Performance Analysis

Figure D.1 shows the performance of our model with varying numbers of VQ embeddings. The results demonstrate that the number of embeddings does not affect model performance. This is expected, as our proposed method primarily focuses on feature dimensionality reduction and clustering. Combined with the overall performance results in Figure C.1, we observe that the model maintains consistent performance.

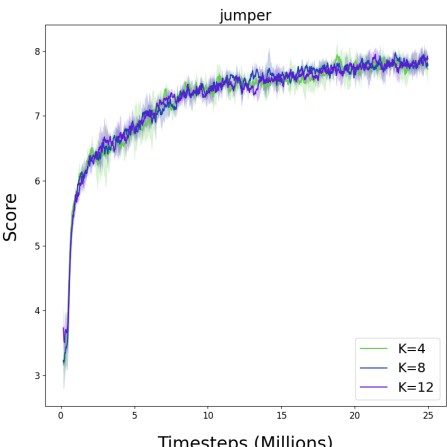

Figure D.1: Performance comparison of models with different numbers of VQ embeddings in the Jumper environment.

## D.2  Interpretability Analysis

Figures D.2a and D.2b illustrate the FDR results for $K = 4$ and $K = 12$, respectively. Our method effectively produces clusters that are clearly separable, regardless of the VQ embedding number.

However, interpretability is influenced by the choice of $K$. When $K = 12$, the semantic clusters become overly fragmented, making it difficult to form coherent semantic explanations for clusters. Conversely, when $K = 4$, Table 3 shows that clusters contain multiple distinct semantic explanations, which negatively impacts interpretability.

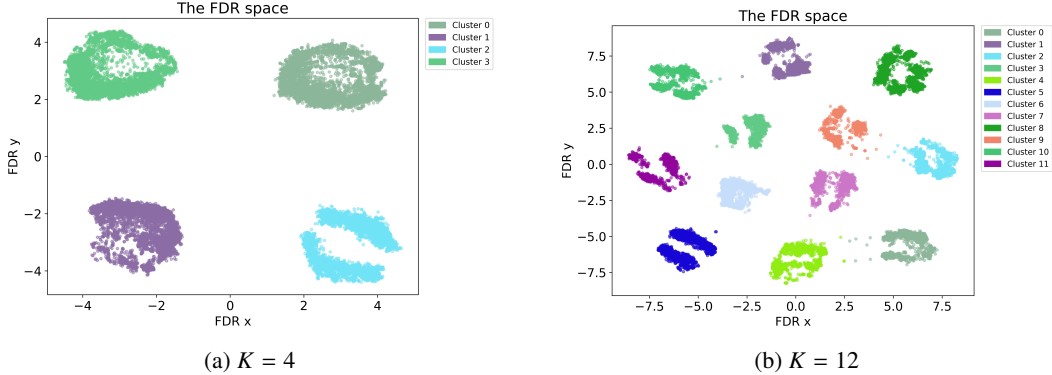

(a) $K = 4$                                        (b) $K = 12$

Figure D.2: Visualization of the FDR spaces for different numbers of VQ embeddings in the Jumper environment.

Clusters with incomplete or incoherent semantic descriptions hinder interpretability by introducing ambiguity in understanding the agent's behavior. This lack of clarity complicates policy analysis and makes it challenging to draw meaningful insights. Conversely, when a single cluster contains multiple interpretable behaviors, it increases the cognitive load for users who must disambiguate

Table 3: Cluster descriptions for the Jumper game with $K = 4$

| Cluster | Description |
|---|---|
| 0 | 1) The agent is touching the carrot on the upper left.
2) The agent is touching the carrot on the right.
3) The agent is touching the carrot on the bottom right.
4) The agent is moving in the left or lower-left part of the scene. |
| 1 | 1) The agent is touching the carrot above.
2) The agent is touching the carrot on the left.
3) The agent is moving in the right or lower-right part of the scene. |
| 2 | 1) The agent is touching the carrot below.
2) The agent is moving in the upper part of the scene. |
| 3 | 1) The agent is moving at the bottom of the scene.
2) The agent is approaching the carrot above. |

between these behaviors. Such a many-to-one mapping between behaviors and clusters undermines the straightforward identification of the agent's current strategy, reducing the utility of clustering as a tool for decision-making. To address these challenges, it is essential to ensure a one-to-one mapping between clusters and explanations. When each cluster is associated with a single, coherent explanation, it eliminates the need for further distinctions within clusters, facilitating clear policy analysis and enhancing human understanding of the agent's behavior.

# E    Clustering Quality under Different Dimensionality-Reduction Methods

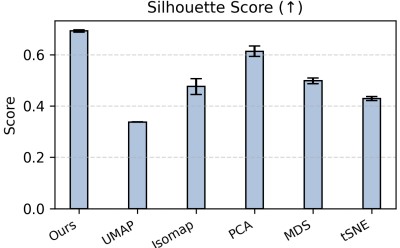
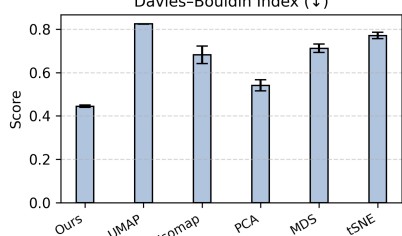

(a) Silhouette score (higher is better)              (b) Davies–Bouldin Index (lower is better)

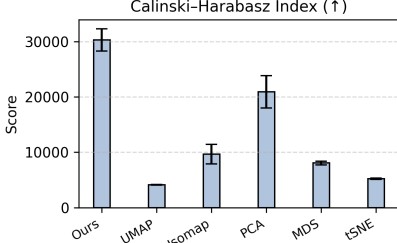

(c) Calinski–Harabasz score (higher is better)

Figure E.1: Clustering metrics averaged over three Procgen games (Ninja, Jumper, and Fruitbot), each run with three random seeds (nine runs in total). Bars show mean values; error bars denote the standard error of the mean.

To quantify how well various dimensionality-reduction (DR) techniques preserve the semantic clusters discovered by our model, we project the same set of high-dimensional state features—collected following the procedure described in § 4.1—using five popular DR baselines: UMAP, Isomap, PCA,

MDS, and t-SNE. These metrics were selected to evaluate both the compactness within clusters and the separation between clusters in the reduced feature space.

- Silhouette – cohesion vs. separation of each point's cluster.
- Davies–Bouldin Index (DBI) – average cluster similarity (lower indicates tighter, more separated clusters).
- Calinski–Harabasz (CH) – ratio of between- to within-cluster dispersion.

Figure E.1 shows that our learned 2-D FDR space achieves the best scores on all metrics: Silhouette is the highest, DBI the lowest, and CH an order of magnitude larger than any baseline. These results confirm that our dimensionality-reduction module preserves—and even sharpens—the intrinsic semantic clustering properties uncovered in the high-dimensional feature space.

# F    Semantic Formation in Clusters

To analyze how semantic clusters form in the feature space, we sample 50,000 states per environment and compute: (i) the per-cluster mean image and the mean (± std) pixel distance from each state to its cluster mean (averaged over $K=8$ clusters), and (ii) the probability of cluster transitions along trajectories. We evaluate three models: *Trained* (ours), *Stop-Grad* (the ablation in § C.2 that removes the effect of our modules by stopping gradients), and *Raw* (untrained). Results are shown in Table 4. Compared to *Raw*, both *Trained* and *Stop-Grad* reduce transition probability, indicating policy-induced structure. However, *Stop-Grad* lacks fully distinct boundaries: it exhibits higher transition probability and lower intra-cluster pixel distance than *Trained*, whereas our method achieves the lowest transition probability and the highest intra-cluster pixel distance.

Table 4: Cluster transition probability and intra-cluster pixel distance (mean with std) over 50k states.

| Environment | Model | Cluster transition probability | Pixel distance mean (Std. Dev.) |
|---|---|---|---|
| FruitBot | Trained | 0.1081 | 100.00 (71.33) |
| FruitBot | Stop-Grad | 0.1520 | 93.21 (68.09) |
| FruitBot | Raw | 0.2834 | 77.10 (49.94) |
| Jumper | Trained | 0.2224 | 110.29 (62.29) |
| Jumper | Stop-Grad | 0.3015 | 108.38 (61.22) |
| Jumper | Raw | 0.5829 | 104.57 (58.31) |
| Ninja | Trained | 0.2680 | 141.57 (67.88) |
| Ninja | Stop-Grad | 0.2705 | 132.46 (66.12) |
| Ninja | Raw | 0.2712 | 87.61 (62.43) |

We further assess temporal coherence with two episode-level metrics: (i) *Episode Cluster Entropy (ECE)*—the entropy of each episode's cluster distribution (lower is better, indicating more focused semantic grouping), and (ii) *Temporal Cluster Agreement* TCA@$k$—the fraction of frame pairs at lag $k$ assigned to the same cluster (higher is better, indicating smoother, more stable semantics). As summarized in Table 5, our method consistently achieves lower ECE and higher TCA@3/6 than *Stop-Grad* across all three games. Note that *Jumper* episodes are shorter, yielding fewer clusters per episode and thus lower ECE values overall.

Table 5: Episode- and frame-wise metrics (averaged over episodes).

| Environment | Model | ECE | TCA@3 | TCA@6 |
|---|---|---|---|---|
| FruitBot | Stop-Grad | 1.9068 | 0.7050 | 0.5309 |
| FruitBot | Ours | 1.8886 | 0.7302 | 0.5502 |
| Jumper | Stop-Grad | 0.6987 | 0.7040 | 0.6715 |
| Jumper | Ours | 0.5572 | 0.7911 | 0.7089 |
| Ninja | Stop-Grad | 1.5505 | 0.5798 | 0.3923 |
| Ninja | Ours | 1.3086 | 0.6787 | 0.4985 |

# G   More Examples and Mean Images in the FDR Space

## G.1   CoinRun

To augment the exploration of semantic clustering as discussed in the main paper, this section analyzes two additional games characterized by distinct dynamics. CoinRun's gameplay mechanism is similar to Ninja's, requiring the agent to traverse from the far-left to the far-right, scoring points by interacting with coins at the far-right end of the scene, as illustrated in Figure G.1.

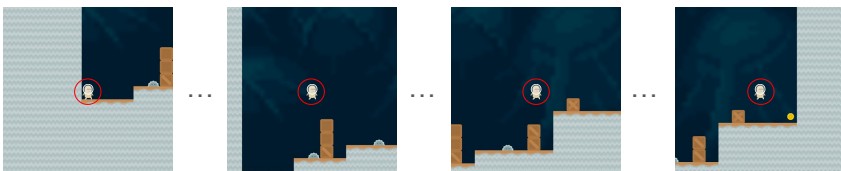

Figure G.1: A episode in CoinRun. Ellipses represent the omitted states.

The observations and insights obtained closely mirror those derived from the analysis of Ninja. Interested readers can leverage the provided code and checkpoint for further exploration of similar findings. Consequently, for brevity, we refrain from extensively elaborating on analogous conclusions.

## G.2   Jumper

In Jumper, the agent navigates a cave to locate and touch carrots by interpreting a radar displayed in the upper right corner of the screen. The radar's pointer indicates the direction of the carrot, while a bar below the radar shows the distance between the agent and the carrot—shorter bars imply closer proximity, and vice-versa.

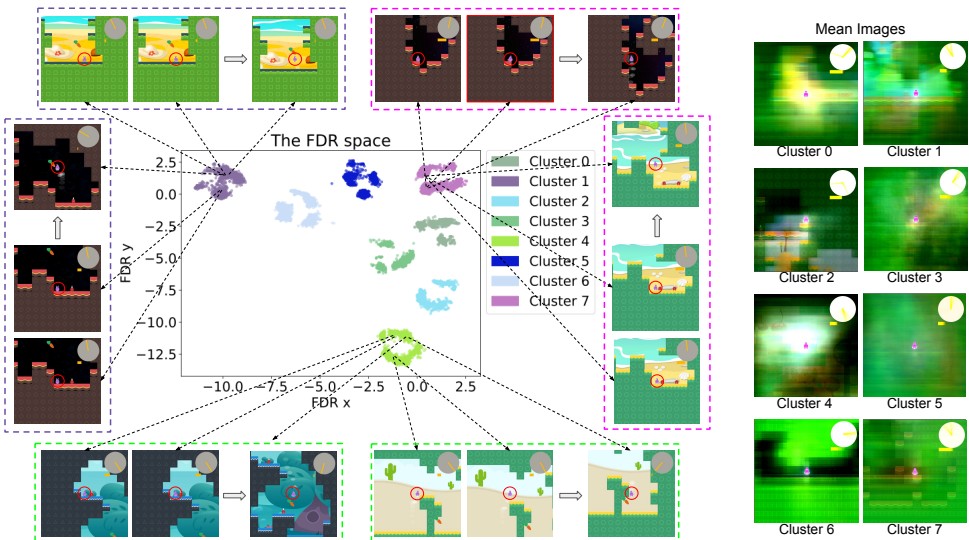

Figure G.2: Examples and mean images from the Jumper FDR space.

The state examples and mean images from the clusters in the FDR space of Jumper are presented in Figure G.2. The background of Jumper is diverse, and the agent is always in the center of the screen (zoom in to see the outlines clearly). In Table 6, we break down descriptions of the sampled images from each cluster and interpretations of the mean image for each cluster in the Jumper game.

Figure G.3 depicts various states from the Jumper game. C.3(a) and C.3(b) belong to the same episode and fall under Cluster 4, while C.3(c) and C.3(d) are from another episode, both categorized under Cluster 1. Notably, neither C.3(a) nor C.3(c) shows the presence of carrots. This observation leads us to suspect that the determination of these clusters is solely reliant on the radar and distance bar

Table 6: Cluster descriptions and mean image outlines for the Jumper game

| Cluster | Description | Mean image outlines |
|---|---|---|
| 0 | The agent learns to jump up from the bottom left and move to the left on the top right. | The radar pointing up and to the right, and the outline of the channel above and to the right.. |
| 1 | The agent is touching the carrot on the left or upper left. | The radar is pointing to the upper left and is very close to the target. |
| 2 | The agent learns the skill of movement at the top of the scene. | The radar pointing to the left or down, and the outline of the channels face to the left or down. |
| 3 | The agent is approaching the carrot on the upper right. | The radar pointing to the upper right, and the distance to the target is very close. |
| 4 | The agent is touching the carrot below. | The radar is pointing down, and it is very close to the target. |
| 5 | The agent is approaching the carrot above or left. | The radar is pointing up, and it is very close to the target. |
| 6 | The agent is touching the carrot on the right. | The radar is pointing right, and it is very close to the target. |
| 7 | The agent learns the skill of movement at the right bottom of the scene. | The radar pointing up or to the top left, and it is far from the target. |

rather than the appearance of carrots. To test this hypothesis, we removed the carrot in C.3(e), which originally belonged to Cluster 4, and transformed it into C.3(f). The result demonstrated that C.3(f) still belongs to Cluster 4, confirming our suspicion. However, this phenomenon might pose potential risks in practical applications. For example, in scenarios where sensor data and visual perceptions misalign, AI models might solely rely on sensor data for decision-making—e.g., an autonomous vehicle's sensors indicating an empty road while the occupants inside observe pedestrians crossing, yet the vehicle continues to accelerate.

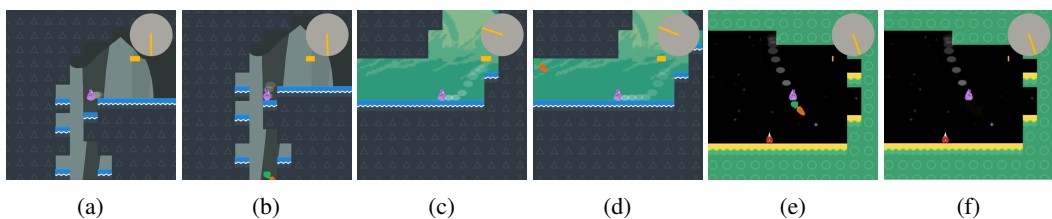

| (a) | (b) | (c) | (d) | (e) | (f) |

Figure G.3: Policy analysis examples in Jumper.

## G.3 FruitBot

FruitBot is a bottom-to-top scrolling game where the agent moves left or right to collect fruits for points while avoiding negative scores upon touching non-fruit objects. The state examples and mean images from the clusters in the FDR space of FruitBot are presented in Figure G.4 and their descriptions in Table 7. FruitBot's mean images lack clarity due to the presence of diverse backgrounds, and the agent is constantly moving to the left and right at the bottom of the screen. However, we can still make out the outline of the wall and agent if we look carefully (zoom in to see the outlines clearly).

We examined a substantial number of video states and corresponding cluster information, and found that the factors determining clusters in FruitBot are the agent's position on the screen and its relative positioning to walls and gaps. This suggests that the agent has learned critical factors within the environment.

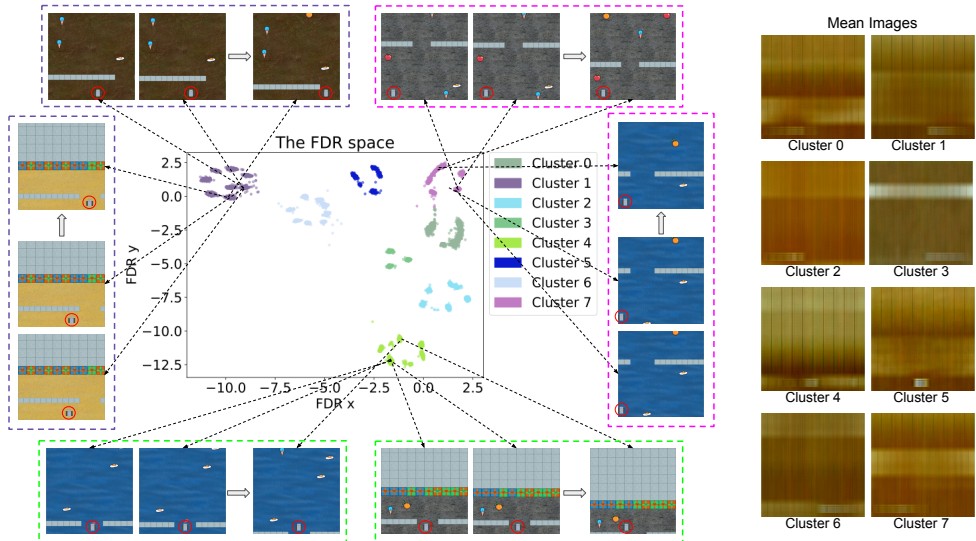

Figure G.4: State examples and mean images from the FruitBot FDR space.

Table 7: Cluster descriptions and mean image outlines for the FruitBot game

| Cluster | Description | Mean image outlines |
|---|---|---|
| 0 | The agent is approaching the wall in the left area. | We can see the agent moving toward the gap on the wall that is approaching on the lower left. |
| 1 | The agent approaches the wall from the right area. | The agent is moving toward the gap on the wall that is approaching on the lower right. |
| 2 | The agent executes its policy far from the wall from the left area. | The wall is far away, and the agent is moving in the lower left. |
| 3 | The agent approaches the wall from the right, but it is still some distance away. | The wall is far away, and the agent is moving in the lower right. |
| 4 | The agent is going through the gap in the middle and left, and insert the key at the end of the scene. | The agent going through the final gap and inserting the key. |
| 5 | The agent approaches the wall from the middle area. | We can identify the outline of the agent in the lower middle. |
| 6 | The agent going through the gap on the right, and performs policy far from the wall in the right area. | The agent is crossing the gap in the bottom right. |
| 7 | The agent approaches the wall from the left, but it is still some distance away. | The agent is moving in the lower left, and the outline of walls is in the middle of the screen. |

# H  Hovering Examples

In figures H.1, and H.2, we present examples of our interactive visualization tool applied to Jumper and FruitBot. This tool is included in the supplementary material, allowing readers to freely explore the semantic distribution of features and gain a better understanding of the semantic clustering properties of DRL.

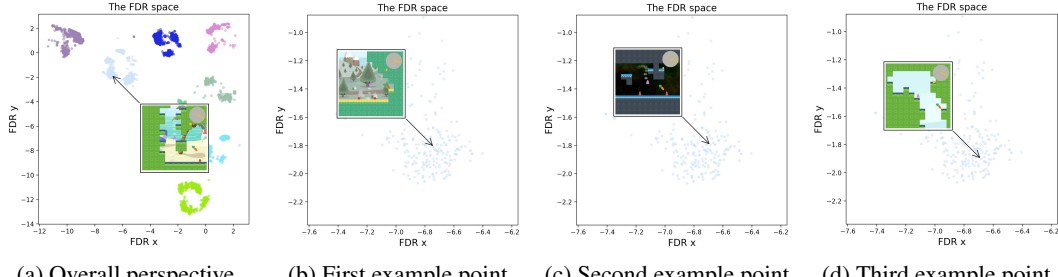

(a) Overall perspective. (b) First example point (c) Second example point (d) Third example point

Figure H.1: Hover examples in the FDR space of Jumper. We observe a sub-cluster in the FDR space as an example from the overall perspective (a) and the zoomed-in perspective (b), (c), and (d). The agent is standing on the edge of a ledge. Although the scenarios of (b), (c), and (d) are different, the proposed method effectively clusters semantically consistent features together in the FDR space.

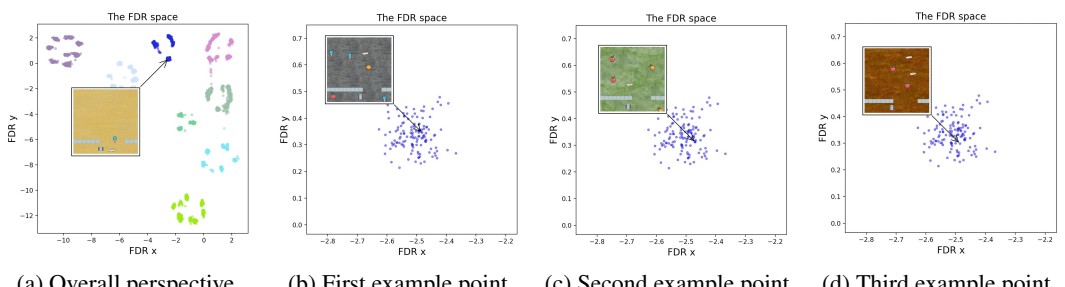

(a) Overall perspective. (b) First example point (c) Second example point (d) Third example point

Figure H.2: Hover examples in the FDR space of Fruitbot. We observe a sub-cluster in the FDR space as an example from the overall perspective (a) and the zoomed-in perspective (b), (c), and (d). The agent is standing on the edge of a ledge. Although the scenarios of (b), (c), and (d) are different, the proposed method effectively clusters semantically consistent features together in the FDR space.

# I Human Evaluation Details

## I.1 Part 1: Overview and Timeline Introduction

This section provides a detailed description of the evaluation process and presents the timeline for conducting the assessment. The evaluation aims to assess the clarity and interpretability of the semantic clusters in the FruitBot, Jumper, and Ninja games, with the objective of enhancing and quantifying the explainability of the DRL system.

### I.1.1 Timeline

Each participant will complete a survey for two game environments (FruitBot, Jumper, or Ninja). They follow the format as detailed:

- **Stage 1: Questionnaire (5 minutes)**
  Participants are requested to complete a questionnaire that collects demographic information and gaming-related details. The questionnaire includes sections for gender, age group, education level, occupation, gaming experience, familiarity with evaluating game states, and preferred game genres.

- **Stage 2: Introduction to Game Environment (10 minutes)**
  During this stage, participants receive an introduction to the evaluation process. They are informed about the objectives of the assessment and the significance of evaluating the clarity and interpretability of the semantic clusters. They are also given a short description of the game environment (FruitBot, Jumper, or Ninja), and are shown a short gameplay clip to aid with the understanding of the game's objectives and features.

- **Stage 3: Assessment (50 minutes)**
  Following the familiarization period, participants spend 50 minutes assessing the semantic

clusters in the game environment. They focus on evaluating the clarity and understandability of the video clips within each semantic cluster. This is done online via a survey.

Total Evaluation Time: 60 minutes.

## I.2    Part 2: Questionnaire

This section of the evaluation plan presents the questionnaire that participants are required to complete. The questionnaire consists of the following sections:

### I.2.1    Demographic Information

- **Age**: Participants indicate their age.
- **Gender**: Participants specify their gender as Male, Female, or Other.
- **Education Level**: Participants indicate their highest level of education completed, including options such as High school and below, Bachelor's degree, Master's degree, and Doctorate and above.
- **Occupation**: Participants provide their current occupation, selecting from options such as Student, Employee, Self-employed, or Other.

### I.2.2    Gaming-related Information

- **Gaming Experience**: Participants indicate their level of gaming experience, choosing from options such as Beginner, Intermediate player, Advanced player, or Professional player.
- **Game Frequency**: Participants indicate their frequency of gaming, choosing from options such as Daily, Several times a week, Weekly, Monthly, or others.
- **Experience in Evaluating Game States**: Participants assess their experience in evaluating game states, selecting from options such as No experience, Some experience, Moderate experience, or Extensive experience.
- **Preferred Game Genres**: Participants specify their preferred game genres, including options such as Role-playing games, Shooting games, Strategy games, Puzzle games, or Other.

## I.3    Part 3: Evaluation Questions for Clarity Assessment

In this section, a comprehensive set of questions is provided to assess the clarity and understandability of the semantic clusters. The questions capture participants' opinions and perceptions using a Likert scale ranging from 'Strongly Disagree' to 'Strongly Agree'. The specific evaluation questions for the clarity assessment include:

- The clips of each cluster consistently display the same skill being performed.
- The clips of each cluster match the given skill description.

The two questions are asked each time the participant has been shown a semantic cluster.

## I.4    Part 4: Evaluation Questions for Interpretability Assessment

This section outlines the question evaluating the interpretability of the semantic clusters in terms of their usefulness. The question is designed to capture participants' opinions and perceptions using a Likert scale ranging from "Strongly Disagree" to "Strongly Agree." The specific evaluation question for the interpretability assessment:

- The identified skills aid in understanding the environment and the AI's decision-making process.

The above question is asked after the participants have seen all the semantic clusters.

### I.5 Part 5: Personnel and Coordination

This section outlines the personnel and coordination aspects of the evaluation. It includes information about evaluator recruitment and compensation. Specifically:

#### I.5.1 Evaluator Recruitment

15 evaluators are recruited to participate in the evaluation.

#### I.5.2 Evaluator Compensation

Each evaluator receives $15 in compensation for their valuable time and contribution to the evaluation process.

By implementing this comprehensive evaluation plan, we gather valuable insights into the clarity and interpretability of the semantic clusters in the FruitBot, Jumper, and Ninja games. The evaluation results provide essential guidance for further quantifying the improved interpretability of DRL models using our proposed method.

### I.6 Part 6: Grouping Details

- **Evaluator 1**: FruitBot, Jumper
- **Evaluator 2**: FruitBot, Jumper
- **Evaluator 3**: FruitBot, Jumper
- **Evaluator 4**: FruitBot, Jumper
- **Evaluator 5**: FruitBot, Jumper
- **Evaluator 6**: Jumper, Ninja
- **Evaluator 7**: Jumper, Ninja
- **Evaluator 8**: Jumper, Ninja
- **Evaluator 9**: Jumper, Ninja
- **Evaluator 10**: Jumper, Ninja
- **Evaluator 11**: Ninja, FruitBot
- **Evaluator 12**: Ninja, FruitBot
- **Evaluator 13**: Ninja, FruitBot
- **Evaluator 14**: Ninja, FruitBot
- **Evaluator 15**: Ninja, FruitBot

This grouping plan ensures that each evaluator evaluates two different games, and each game receives a total of 10 evaluations. It allows for comprehensive evaluations of each game and ensures that evaluators have an opportunity to provide feedback on multiple games.

## J Potential Societal Impacts

This paper advances the interpretability of DRL through semantic clustering, with potential applications in safety-critical domains such as autonomous systems and robotics. While primarily contributing to the field of Machine Learning, we encourage responsible application to mitigate potential misuse and do not identify immediate societal or ethical risks requiring specific emphasis.

