# OpenReview forum: "Enhancing Interpretability in Deep Reinforcement Learning through Semantic Clustering"
_NeurIPS.cc/2025/Conference — NeurIPS 2025 poster_

### Official Review · Reviewer_fcAv · 2025-06-22

**Clarity:** 3
**Significance:** 3
**Originality:** 3
**Rating:** 5
**Confidence:** 2

**Summary:**

The paper proposes a plug-in “semantic clustering module” for deep-reinforcement-learning (DRL) agents that pairs a lightweight 2-D feature-dimensionality-reduction (FDR) network with an online vector-quantiser. Jointly trained with any standard DRL algorithm (shown with PPO on the Procgen benchmark), the module yields a stable, human-interpretable partition of the latent state space into discrete clusters without hurting task performance. These clusters expose the structure of learned policies, enable video-level explanations such as “approaching the right-wall then jumping to a higher ledge,” and let practitioners inspect or segment roll-outs by semantic “skills.” Experiments across CoinRun, Ninja, Jumper and FruitBot demonstrate markedly clearer, seed-independent clusters than t-SNE visualisations, high agreement in a 15-person user study (Likert ≈ 4.2/5), and no significant score drop versus baseline PPO. Overall, the work shows that DRL already forms meaningful semantic groupings internally and that making them explicit dramatically improves transparency while remaining computationally light.

**Questions:**

* Limited discussion on the benefits of adding vector codes to the state features

The authors state that “Training the VQ code $k$ with a latent-conditioned policy $\pi(a∣s,k)$ supports the extension to downstream tasks, such as macro action selection in hierarchical learning.” However, this appears to be the only justification provided for incorporating the VQ code into the state features. Given that the paper does not include any experiments in a hierarchical learning setting, nor an ablation study comparing performance with and without the VQ-code, this design choice feels insufficiently motivated. Since the main goal of the paper is to cluster state features—rather than to influence the policy—further discussion is warranted. An ablation study examining the effect of excluding the VQ-code could significantly enhance the clarity and rigor of the work.

* Inconsistency between Section 4.1 and Appendix C.2

In Section 4.1, the authors state:
“The t-SNE visualization of PPO (Figure 2a), spreads features across the space without forming clear clusters, limiting its utility for clustering analysis and requiring detailed manual examination of certain areas, as in previous studies.”

However, in Appendix C.2, the “stop-gradient” experiment shows:
“The results, as shown in Figures C.2 and C.3, demonstrated that states within the same semantic cluster continued to exhibit similar semantic interpretations, even without the influence of the semantic clustering module.”

This comparison seems somewhat inconsistent. If I understand correctly, the policy and the state representations in vanilla PPO and the stop-gradient method should be identical, assuming all other components remain the same. Figure C.2 in the appendix shows a t-SNE visualization of the stop-gradient method's feature space with cluster labels—yet these clusters should be equivalent to those from vanilla PPO, as the policy is unaffected. In that case, the two quoted statements are not strictly contradictory, but they do feel awkward when read together. The apparent discrepancy seems largely due to the absence of color labeling in Figure 2, which may lead to a misleading interpretation.

I recommend revising Section 4.1 to explicitly incorporate and contrast the stop-gradient experiment, clarifying how your method differs from it and why joint training offers benefits. Additionally, it would be helpful to include an *unlabeled* version of your method's t-SNE plot, to mirror the presentation of vanilla PPO more symmetrically. This would provide a fairer and clearer comparison of visual cluster separation and further support your narrative.


Furthermore, if the main difference between your method and the stop-gradient baseline lies in sharper cluster boundaries, then this distinction deserves richer discussion why is this important. Your analysis operates on the clusters themselves, not directly on the FDR or t-SNE visualizations (which shows sharper bounds). More evidence is needed to support the claim that sharper boundaries improve semantic grouping. For instance, examples of ambiguous or misclustered states at the boundaries would be valuable.

Lastly, the Semantic Formation experiment in Appendix F is not fully convincing. The statistics are averaged over 50,000 steps; it would be more informative to show episode-level metrics or frame-wise agreement scores. A comparison of Semantic Formation between your method and the stop-gradient baseline could help demonstrate the added value of your design. If your method indeed shows higher consistency or interpretability on these metrics, that would substantially strengthen your claims.

* Limited performance comparison and sensitivity analysis of the newly introduced hyper-parameters

Performance comparisons are limited to a small set of Procgen environments. Evaluation on additional domains—such as Atari or other standard DRL benchmarks—would improve the generalizability of your claims.

It is commendable that your method preserves performance without requiring changes to the original PPO hyper-parameters and also demonstrates robustness to random seeds—both of which are important properties in the reinforcement learning domain and provides study of the effect of K. However, the method introduces new hyper-parameters (e.g., weights for the semantic loss components), and their sensitivity is not thoroughly explored. Understanding how these additional parameters affect performance and interpretability is crucial for assessing the stability and general usability of the approach. A more detailed sensitivity analysis would greatly strengthen the paper's practical value and reproducibility.

**Ethical Concerns:**

["NO or VERY MINOR ethics concerns only"]

**Final Justification:**

The authors have addressed my questions and concerns regarding their method and the presented ablations. The additional metrics on semantic focus and temporal coherence make their claims more compelling and better substantiated.

Overall, I found the work interesting and potentially impactful. It could inspire a new line of research, especially when combined with automatic state description generation using modern vision-language models—a direction I find particularly intriguing.

**Limitations:**

yes

**Quality:**

3

**Strengths And Weaknesses:**

**Strengths:**

* Novel approach to the important task of interpretable reinforcement learning
* Strong and versatile analysis of the proposed method
* Rigorous human study (though limited for making stronger behavioral claims)
* Useful visualization tool

**Weaknesses** (see detailed explanations in the questions):

* Limited discussion on the benefits of adding vector codes to the state features
* Inconsistency between Section 4.1 and Appendix C.2
* Limited performance comparison and analysis of the sensitivity to the newly introduced hyper-parameter
* Additional weaknesses noted in the discussion: choice of *k*, lack of automated description generation, etc.

In essence, this is a strong paper with several weaknesses that, however, do not undermine the overall contribution.

---

> ### Author Rebuttal · Authors · 2025-07-28
>
> # Rebuttal to Reviewer fcAv
>
> We sincerely appreciate Reviewer fcAv for their encouraging evaluation and for highlighting our novel method, analyses, user study, and visualization tool. Below, we address each concern.
>
> ---
>
> ## Questions & Weaknesses
>
> > 1. Limited discussion on the benefits of adding vector codes to the state features
>
> Our experiments confirm that integrating the VQ code into policy training has negligible impact on model performance (see lines 153–154, 475–476; App. C.1, D), while lines 226–231 presents how the VQ code aids analysis of hierarchical policy structures and improves interpretability in downstream hierarchical learning.
>
> ---
>
> > 2. Inconsistency between Section 4.1 and Appendix C.2
>
> 1) **Consistency Clarification**
>    As stated the captions of Fig 2 and Fig C.2, cluster colors (cluster indices) in the t-SNE visualizations are derived from our proposed method and are used solely to facilitate comparison of spatial relationships. This note will also be added to the main text to prevent misinterpretation in the final version.
>
> 2) **Discussion of Stop-Gradient Baseline**
>    In Section 4.1, we will introduce a discussion about the stop-gradient experiment in App C.2, highlighting how our approach yields sharper and more coherent clusters.
>
> 3) **Boundary-State Ambiguities**
>    In the stop-gradient t‑SNE (Fig. C.2), overlapping cluster regions cause semantically similar states to be split across clusters, weakening semantic separation (see lines 473–475). We will include concrete examples of these boundary-state ambiguities in the final version.
>
> 4) **Semantic Formation Comparison & New Metrics**
>    As suggested, we added two experiments that confirmed our method both improves semantic focus and strengthens temporal coherence.
>
>    - **Experiment 1: Stop‑Grad Baseline**
>      We extended Table 4 to include the Stop‑Grad baseline:
>
>      | Environment  | Model      | Cluster transition probability | Pixel distance mean (Std Dev.) |
>      |:------------:|:-----------|:------------------------------:|:------------------------------:|
>      | **FruitBot** | Trained       | 0.1081                         | 100.00 (71.33)                  |
>      |              | Stop‑Grad  | 0.1520                         | 93.21 (68.09)                   |
>      |              | Raw        | 0.2834                         | 77.10 (49.94)                   |
>      | **Jumper**   | Trained       | 0.2224                         | 110.29 (62.29)                  |
>      |              | Stop‑Grad  | 0.3015                         | 108.38 (61.22)                  |
>      |              | Raw        | 0.5829                         | 104.57 (58.31)                  |
>      | **Ninja**    | Trained       | 0.2680                         | 141.57 (67.88)                  |
>      |              | Stop‑Grad  | 0.2705                         | 132.46 (66.12)                  |
>      |              | Raw        | 0.2712                         | 87.61 (62.43)                   |
>
>      *Analysis:* Stop‑Grad preserves the policy‑induced semantic grouping but lacks fully distinct boundaries, resulting in a higher cluster transition probability and a lower intra‑cluster pixel distance compared to our method (Trained).
>
>    - **Experiment 2: Episode‑ and Frame‑wise Metrics**
>      We tested two more metrics:
>      - **Episode Cluster Entropy (ECE):** entropy of the episode’s cluster distribution (lower → more focused semantic grouping).
>      - **Temporal Cluster Agreement (TCA@k):** fraction of frame pairs at lag k sharing the same cluster (higher → smoother, more stable semantics).
>
>      | Environment  | Model      | ECE    | TCA@3  | TCA@6 |
>      |:------------:|:-----------|:-------|:-------|:-------|
>      | **FruitBot** | Stop‑Grad  | 1.9068 | 0.7050 | 0.5309 |
>      |              | Ours       | 1.8886 | 0.7302 | 0.5502 |
>      | **Jumper**   | Stop‑Grad  | 0.6987 | 0.7040 | 0.6715 |
>      |              | Ours       | 0.5572 | 0.7911 | 0.7089 |
>      | **Ninja**    | Stop‑Grad  | 1.5505 | 0.5798 | 0.3923 |
>      |              | Ours       | 1.3086 | 0.6787 | 0.4985 |
>
>      *Analysis:* Across all three games, our method achieves lower ECE (indicating more concentrated clusters) and higher TCA@3/6 (indicating stronger temporal coherence) compared to Stop‑Grad. Note that Jumper episodes are shorter than those in the other two games, which presents fewer clusters in single episodes and therefore yields lower ECEs.
>
>    - We will include these results in Appendix F of the final version.
>
> ---
>
> > 3. Limited performance comparison and sensitivity analysis of newly introduced hyper-parameters
>
> The primary goal of this paper is to enhance DRL interpretability (lines 39–48). While not the focus, we include performance analysis throughout: 1) Appendix C provides ablation results; 2) lines 406–407 outline hyperparameter selection; 3) Appendix D examines how varying the number of VQ embeddings affects both performance and interpretability.
>
> During hyper-parameter tuning, we found performance is mainly influenced by $w_{FDR}$, $w_{VQ\text{-}VAE}$, and $f_{control}$, and is robust to the number of VQ embeddings and the degrees of freedom in Eq. (3). We will report this finding in the final version. Furthermore, we plan to expand benchmarks (lines 252–253) and provide further performance analysis in future work. We also commit to releasing our code upon paper acceptance, enabling the community to replicate and extend our results.

---

> > ### Comment · Reviewer_fcAv · 2025-08-04
> > **Acknowledging the Rebuttal**
> >
> > I thank the authors for their rebuttal and for addressing the concerns raised in the initial review. I especially appreciate the additional experiments and clarifications, which in my opinion provide more adequate evidence in support of the paper’s claims.
> >
> > I will keep my score.

---

> > > ### Author Response · Authors · 2025-08-05
> > >
> > > Thank you for reviewing our rebuttal and for your thoughtful feedback. We’re glad the additional experiments and clarifications strengthened our claims, and we truly appreciate your continued evaluation and support.

---

### Official Review · Reviewer_AFfc · 2025-07-03

**Clarity:** 2
**Significance:** 3
**Originality:** 3
**Rating:** 5
**Confidence:** 3

**Summary:**

This paper investigates methods for interpreting policies in deep reinforcement learning. In particular, they propose a method to cluster states used by the policy in a way that respects semantic differences in the policy behaviour. There are a few qualitative investigations in ProcGen, with the rationale that policies in procedural generation would face a generalization challenge where interpretability would be useful for understanding the learned policy.

**Questions:**

- Figure 1: Why is it that the VQ code is integrated into the feature, rather than the VQ embedding?
- Section 3.1 (curse of dimensionality): Wouldn't the curse of dimensionality also apply to the problem of dimension reduction? I am not sure I understand this motivation if the VQ code needs to be learned.
- Section 3.1 (distance relationships): Would the reduced features learned from preserving consistency across distance relationships also serve to represent relationships with respect to other distances, like bisimulation metrics?
- Section 3.2 (FDR loss): It is not clear to me what the relationship is between p and q here, and why the loss takes the form that it does. By optimizing the FDR loss, are you ensuring that the similarities of induced by f and g \circ f are similar?
- Section 3.2 (vector magnitude in student T): I understand that using the same $\alpha$ ensures the same distance relationship, but wouldn't the magnitudes of the vectors differ if f and g \circ f are of different dimensions?
- Section 3.2 (Modified VQ-VAE): without the third term, how does the encoder of the VQ-VAE learn to maintain the assigned embedding? I understand that the $e_k$ can cover the space as centroids of K means, but an important difference is that the underlying features are also changing as the encoder is learned.
- Section 4 (Cluster separation and t-sne): I wonder what exactly the t-sne is capturing on top of the learned clusters of the embeddings. Does t-sne on the fused state feature essentially return the cluster index?
- Section 4 (sensitivity to number of states/seed): While the overall structure of the clustering remains the same, I wonder if there is any way in which the identity of these clusters could be identified? For example, does the specific embedding (e_k) assigned to these clusters change?
- Figure 3 and Table 1: The results presented here are a bit difficult to follow because the change in subsequent images is difficult to detect. For example, cluster 4 which is described as performing a jump has mostly images of the agent near the top of its jump rather than a scene closer to the ground at the beginning of the jump. I am also not sure if the mean images accurately depict the dynamic nature.
- Table 2: While these results show the proposed method in a favourable light, they would be much stronger if the evaluation included clips from clusters of a baseline method for comparison.
- Section 5: an important limitation that was left out is whether the method has any effect on performance. I understand that this not the primary concern of this paper, but training the policy with the proposed fused features could result in changes to performance.

### Minor Comments
- Equation 2: I suggest using another symbol for the "control factor", since $f$ is usually used to denote a function
- Section 3.2 (Improved Clustering): the first paragraph feels out of place, and i think that it can be demonstrated in the experiments. It doesn't connect with the actual description of the method earlier in the section. THe second paragraph is interesting, but difficult to follow. Could the statement be made as a formal proposition or theorem?
- Figure 2: the titles of the subfigures could be more informative or removed.
- Figure 3: the mean images could benefit increased contrast to highlight the difference between the clusters.

**Ethical Concerns:**

["NO or VERY MINOR ethics concerns only"]

**Final Justification:**

The authors have clarified the questions I raised in my rebuttal, including further justification of including the VQ-code and omitting a term from the VQ-VAE loss, as well as demonstrating that the performance impact is minimal.

**Limitations:**

yes

**Quality:**

3

**Strengths And Weaknesses:**

### Strengths
- The method appears to be generally applicable, allowing it to be added to a conventional RL algorithm without any change to its operation. (Although, I do have a question about whether it can change performance/learning, see below).

- Qualitative analysis suggests that the proposed approach for clustering in drl is generally effective. The experiments seem to be well-designed, thought-out and provide compelling evidence for improvements over t-sne when it comes to interpreting the policy's behaviour semantically.

### Weaknesses
- The presentation of the experimental results could be improved (see below).

- Unclear whether if there is an effect of the proposed feature extraction method on downstream performance. If so, then this is a weakness without an investigation to demonstrate that the performance impact is negligible.

---

> ### Author Rebuttal · Authors · 2025-07-28
>
> # Rebuttal to Reviewer AFfc
>
> We sincerely appreciate Reviewer AFfc for their thoughtful review and for recognizing the general applicability of our method, the quality of the qualitative analyses, and the usefulness of the visualization tool. We address each concern below.
>
> ---
>
> ## Weaknesses
>
> > 1. The presentation of the experimental results could be improved (see below).
>
> Please refer to our response to the questions below.
>
> > 2. Unclear whether the proposed feature extraction affects downstream performance. If so, this is a weakness without an investigation showing negligible impact.
>
> Sections C.1, D.1, and lines 153–154 report that our proposed module has minimal impact on policy performance with experimental evidence.
>
> ---
>
> ## Questions
>
> > 1. Figure 1: Why integrate the VQ code into the feature rather than the VQ embedding?
>
> As stated in lines 140–142, integrating the VQ code into policy training aligns with the definition of a latent-conditioned policy, facilitating downstream task integration. Furthermore, broadcasting the 1-D VQ code and adding it to the state feature is simpler than integrating a multi-dimensional VQ embedding with mismatched dimensions.
>
> > 2. Section 3.1 (curse of dimensionality): Wouldn't the curse of dimensionality also apply to the problem of dimension reduction? I am not sure I understand this motivation if the VQ code needs to be learned.
>
> We avoid clustering in the original high‑D feature space by first compressing features into a low‑D manifold with the FDR network (lines 84–96) and then learning VQ codes on that reduced manifold for clustering (lines 97–100, 118–122), thereby mitigating sparsity and sidestepping the curse of dimensionality.
>
>
> > 3. Section 3.1 (distance relationships): Would the reduced features learned from preserving consistency across distance relationships also serve to represent relationships with respect to other distances, like bisimulation metrics?
>
> FDR optimizes over pairwise similarities; in principle, replacing this affinity with one induced by a bisimulation metric would extend the objective. We will note this as a future direction in §5.
>
> > 4. Section 3.2 (FDR loss): It is not clear to me what the relationship is between p and q here, and why the loss takes the form that it does. By optimizing the FDR loss, are you ensuring that the similarities of induced by f and g \circ f are similar?
>
> $p$ and $q$ are pairwise similarities in the high‑D and low‑D spaces, respectively (lines 110–112). Appendix B.1 derives that minimizing the FDR loss encourages that the low‑dimensional mapping preserves the original similarities.
>
> > 5. Section 3.2 (vector magnitude in student T): I understand that using the same α ensures the same distance relationship, but wouldn't the magnitudes of the vectors differ if f and g \circ f are of different dimensions?
>
> Magnitudes can differ. However, Appendix B.1 formally proves that by fixing the Student’s t degrees-of-freedom in both spaces, all pairwise affinities become proportional—identical up to a positive scale factor—thereby preserving the relative ordering of distances.
>
> > 6. Section 3.2 (Modified VQ-VAE): without the third term, how does the encoder of the VQ-VAE learn to maintain the assigned embedding? I understand that the $e_k$ can cover the space as centroids of K means, but an important difference is that the underlying features are also changing as the encoder is learned.
>
> Appendix B.2 proves that our modified VQ‑VAE is equivalent to online k‑means. The update mechanism ensures centroids continually track the moving feature distribution, yielding stable, semantically coherent clusters despite feature drift. Our experiments demonstrate the effectiveness and stability of this clustering method (see Sec. 4.1).
>
> > 7. Section 4 (Cluster separation and t-sne): I wonder what exactly the t-sne is capturing on top of the learned clusters of the embeddings. Does t-sne on the fused state feature essentially return the cluster index?
>
> We use t‑SNE only as a 2D visualization baseline. Cluster colors (cluster indexes) in the t‑SNE plots are derived from our method to compare spatial relations; t‑SNE does not “recover” indices but provides a layout for visual inspection (see Sec. 4.1; Fig. 2 caption).
>
> > 8. Section 4 (sensitivity to number of states/seed): While the overall structure of the clustering remains the same, I wonder if there is any way in which the identity of these clusters could be identified? For example, does the specific embedding (e_k) assigned to these clusters change?
>
> After training, the learned codebook is fixed at inference. Each cluster maintains a consistent identity tied to its learned codebook vector $e_k$.
>
> > 9. Figure 3 and Table 1: The results presented here are a bit difficult to follow because the change in subsequent images is difficult to detect. For example, cluster 4 which is described as performing a jump has mostly images of the agent near the top of its jump rather than a scene closer to the ground at the beginning of the jump. I am also not sure if the mean images accurately depict the dynamic nature.
>
> Cluster 4’s description refers to the in-air phase (“After performing a high jump, the agent loses sight of the ledge below” in Table 1), so the mean image at apex accurately captures this behavior. Static mean images provide a statistical summary; to illustrate true motion, we include dynamic video clips for every cluster shown in the paper in the supplementary material (lines 191–192). Upon acceptance, we will make these video clips and our code publicly available to facilitate reader understanding.
>
>
> > 10. Table 2: While these results show the proposed method in a favourable light, they would be much stronger if the evaluation included clips from clusters of a baseline method for comparison.
>
> Thank you for the suggestion. Evaluating interpretability baselines (Mnih et al. [17]; Zahavy et al. [32]) is very challenging: t-SNE’s stochasticity yields non-reproducible clusters, and prior methods required extensive manual annotation of only a few large clusters, leaving many smaller clusters unexplained (lines 36–38).
>
> As is standard in interpretability research, we rely on human evaluation (lines 203–219) rather than a baseline comparison (Doshi‑Velez & Kim, *Toward a Rigorous Science of Interpretable ML*, 2017). Table 2’s high scores confirm our method’s interpretability effectiveness. Meanwhile, Appendix E provides a quantitative clustering-quality comparison against PCA, UMAP, t-SNE, etc., underscoring our clear advantage.
>
> > 11. Section 5: an important limitation that was left out is whether the method has any effect on performance. I understand that this not the primary concern of this paper, but training the policy with the proposed fused features could result in changes to performance.
>
> As noted earlier, our method has minimal impact on policy performance (see sections C.1, D.1, and lines 153–154).
>
> ---
>
> ## Minor Revisions
> We are very grateful for the suggestions and will incorporate them to improve clarity of the paper.

---

### Official Review · Reviewer_P422 · 2025-07-03

**Clarity:** 3
**Significance:** 3
**Originality:** 3
**Rating:** 4
**Confidence:** 4

**Summary:**

The authors employ the following argument structure to motivate their paper:

1. t-SNE has been used in DRL works to visualize the encodings of DRL agents
2. t-SNE is unstable (sensitive to initialization and therefore has inconsistent output)
3. t-SNE requires extensive manual cluster annotation
4. It would be good to provide an alternative to t-SNE without the issues identified in 2 and 3

Their solution is to integrate (online) clustering into DRL training. Their solution is built on top of VQ-VAE.

**Questions:**

- I found the sentence on line 107 beginning with "We use the Student's t-distribution..." unclear. What do you mean that it captures nonlinear structures?
- What norm is being used in Equation (3) on line 110?
- On Line 115-116 the authors say that "the original distance relationship between features is maintained in the low-dimensional space." Is it a well-known or easy result that the student's t kernel is an isometry? If not, there should be a citation or proof for this in the appendix.
- On line 127 did you mean to say that the clusters themselves are made more compact?
- What are the qualifications of the human evaluators? I would have appreciated if this information was present in the main body of the paper near line 204-205.
- How is the number of clusters controlled? Is it possible that there are overwhelmingly many for a human to interpret, and if so, what is to be done?

**Ethical Concerns:**

["NO or VERY MINOR ethics concerns only"]

**Final Justification:**

The authors have responded to my concerns, the most significant of which related to substantiating their claim that prior methods require extensive manual annotation while the proposed method does not. I do not view this concern as fully resolved as I would need very detailed comparisons between the proposed method and more than just one or two prior works for such a bold, general claim.

**Limitations:**

The authors did not try to recover an inner product in their embedding space whose quantification of similarity corresponds to semantic similarity in the input space.

**Quality:**

3

**Strengths And Weaknesses:**

## Strengths

- My own prior experiences using t-SNE to visualize DRL agent encodings left me desiring an alternative, so I am glad to see this paper propose one.
- I think the authors successfully addressed the issue of t-SNE being unstable
- Section 4.1 gives some evidence that their method yields more compact and separated clusters than t-SNE does.
- Section 4.1 gives evidence that their method is not unstable like t-SNE is.
- The authors provide a human evaluation. I think these are important for an XAI paper to be convincing.
- I appreciate that the authors developed a visualization tool for analyzing the embedding space of DRL models. This is a nice addition that they didn't necessarily need to add. I liked the policy analysis at the end of section 4. I could see myself using it in the future if it is integrated into some standard python RL library.

## Weaknesses

- Nitpick: I would prefer if the authors gave a definition of "semantic clustering" that does not use the word "semantic" in it (unless the word "semantic" is also given a formal non-self-referential definition).

- I would have appreciated if the **Online Clustering** paragraph in Section 3.1 did more to convince readers that previous work requires "extensive manual annotation." The key word for me there is "extensive." That should be substantiated. I view this as a significant omission.

- I would have appreciated an explanation of Equation (4) rather than just showing the equation and ending the paragraph.

---

> ### Author Rebuttal · Authors · 2025-07-28
>
> # Rebuttal to Reviewer P422
>
> We sincerely thank Reviewer P422 for their thoughtful feedback and for highlighting our alternative to t‑SNE, the stability and compactness of our clusters, the human evaluation, and the visualization tool. Below, we address each concern.
>
> ---
>
> ## Weaknesses
>
> > 1. Nitpick: I would prefer if the authors gave a definition of "semantic clustering" that does not use the word "semantic" in it (unless the word "semantic" is also given a formal non-self-referential definition).
>
> We will revise the definition (lines 19–20) to state that clusters group states eliciting similar agent behaviors under comparable environmental conditions (e.g., approaching a target, jumping to a higher platform in Procgen), avoiding self‑reference.
>
> > 2. I would have appreciated if the Online Clustering paragraph in Section 3.1 did more to convince readers that previous work requires "extensive manual annotation." The key word for me there is "extensive." That should be substantiated. I view this as a significant omission.
>
> We will expand §3.1 to state more details. In our setting, annotators labeled clusters for one environment by reviewing a handful of short clips in **~15 minutes**, whereas manual per‑state feature analysis in earlier approaches without automated clustering (e.g., Mnih et al. [17], Zahavy et al. [32]) typically takes **hours**.
>
> >3. I would have appreciated an explanation of Equation (4) rather than just showing the equation and ending the paragraph.
>
> We will add:  “Minimizing the Eq. (4) encourages the low‑dimensional mapping to preserve the pairwise neighborhood structure of the high‑dimensional features.”
>
> ---
>
> ## Questions
>
> > 1. I found the sentence on line 107 beginning with "We use the Student’s t-distribution..." unclear. What do you mean that it captures nonlinear structures?
>
> By “nonlinear structures,” we refer to curved or folded manifolds in feature space. The Student’s t affinity preserves such manifolds after projection—unlike linear methods (e.g., PCA).
>
> > 2. What norm is being used in Equation (3) on line 110?
>
> L₂. We will clarify this in the final version.
>
> > 3. On lines 115–116 the authors say that "the original distance relationship between features is maintained in the low-dimensional space." Is it a well-known or easy result that the Student’s t kernel is an isometry? If not, there should be a citation or proof for this in the appendix.
>
> We do not claim an exact isometry. Appendix B.1 formally proves that—when using the same degrees-of-freedom parameter in both spaces—the Student’s t affinity in the 2D embedding equals the original affinity up to a positive scale factor.
>
> > 4. On line 127 did you mean to say that the clusters themselves are made more compact?
>
> Yes. We will rephrase to “each cluster become more compact (smaller intra‑cluster distances)...” to improve clarity.
>
> > 5. What are the qualifications of the human evaluators? I would have appreciated if this information was present in the main body of the paper near lines 204–205.
>
> We will add near lines 204–205: annotators were adults (18+), native or highly proficient English speakers, with basic video‑gaming experience and a brief training session.
>
> > 6. How is the number of clusters controlled? Is it possible that there are overwhelmingly many for a human to interpret, and if so, what is to be done?
>
> The number of clusters $K$ is a fixed hyperparameter (see L247–251, Appendix D). Very large $K$ fragments episodes into tiny segments, hindering complete description (lines 499–500).

---

> > ### Comment · Reviewer_P422 · 2025-08-07
> >
> > Thanks for your reply. I am satisfied that most of my points have been addressed.
> >
> > However, regarding my 2nd listed weakness and your reply:
> > > We will expand §3.1 to state more details. In our setting, annotators labeled clusters for one environment by reviewing a handful of short clips in ~15 minutes, whereas manual per‑state feature analysis in earlier approaches without automated clustering (e.g., Mnih et al. [17], Zahavy et al. [32]) typically takes hours.
> >
> > As I mentioned this is a significant omission so I would appreciate if the authors could give more details here so that the reviewers may be convinced that your method requires significantly less manual annotation. In particular, are the annotation tasks for your short clips and the ones in Mnih et al. and Zahavy et al. of comparable difficulty? Are you willing to state a claim that your annotation process *generally* takes on the order of (tens) of minutes while the annotation processes of previous methods take on the order of hours? And if so, what justification and evidence do you have for that claim?

---

> > > ### Author Response · Authors · 2025-08-07
> > >
> > > Thank you for your thoughtful follow-up. We're glad to hear that most of your concerns have been addressed. Regarding your second listed weakness, we appreciate the opportunity to provide additional clarification below.
> > >
> > > ---
> > >
> > > In prior work such as Zahavy et al. (2016), annotating states involved the following steps:
> > >
> > > - **Step 1:** Manually inspect individual states to group semantically similar ones.
> > > - **Step 2:** Review state sequences within and across groups to refine cluster boundaries.
> > > - **Step 3:** Write semantic summaries for each cluster.
> > >
> > > This process is highly labor-intensive, especially since t-SNE visualizations used in prior work (e.g., Figs. 3 in Zahavy et al. (2016)) lack clear cluster boundaries and often separate semantically similar states into disconnected regions, which makes manual grouping even more challenging. Using our hover tool (Fig. 5) with the t-SNE visualizations, we found that examining states in a single cluster (i.e., steps 1 and 2) took more than **20 minutes**. For environments with 8 clusters—as in our experiments in Section 4—annotation would take at least **~2.5 hours per environment**.
> > >
> > > In contrast, our method automatically assigns states into semantically meaningful clusters, eliminating the need for Step 1 and Step 2 (lines 97–100, 118–122). Our annotators watch video clips of consecutive state sequences (e.g., those in the supplementary materials) and provide semantic summary per cluster (e.g., as shown in Table 1). The entire process typically takes just **~15 minutes per environment**.
> > >
> > > We will expand the *Online Clustering* paragraph in Section 3.1 to include the above discussion and would be happy to clarify further if needed. We sincerely thank you for your continued evaluation and thoughtful support in improving our paper.
> > >
> > > ---
> > >
> > > ## References
> > >
> > > - Zahavy, et al., Graying the black box: Understanding DQN, ICML 2016

---

> > > > ### Comment · Reviewer_P422 · 2025-08-08
> > > >
> > > > Thank you for the additional details. I will be keeping my scores the same.

---

> > > > > ### Author Response · Authors · 2025-08-08
> > > > >
> > > > > Thank you for your follow-up and for reviewing our additional details. We truly appreciate the time and effort you have dedicated to evaluating our work.

---

### Official Review · Reviewer_cNxc · 2025-07-03

**Clarity:** 1
**Significance:** 1
**Originality:** 4
**Rating:** 4
**Confidence:** 3

**Summary:**

The work addresses the interpretability aspect of reinforcement learning, by developing a semantic clustering approach, which is trained end to end while learning a new policy. To do this the authors use a VQ-VAE inspired novel architecture quantaization to obtain low dimensional state embeddings. Through experiments on Procgen game environments, they demonstrate that DRL models naturally group states representing similar behaviors or "skills" together, such as "jumping to higher platforms" or "approaching targets," without requiring external supervision.

**Questions:**

1) “cluster inputs based on their semantic similarity in the internal space”: what is the internal space?
2) Why element wise addition of the feature vector? why not concatenate it?
3) If the goal is to analyse the DRL policy, then why make it conditional by adding to the policy term? What if the policy degrades?
4) The results show that sematic similar states are clusterd together. But what if the features which procgen is changing (background), is ignored by the feature generator already (before the clustering module)? Implying the added VQ-VAE is not necessary.

**Ethical Concerns:**

["NO or VERY MINOR ethics concerns only"]

**Final Justification:**

The authors have addressed most of my concerns which were raised during the reviewing process (though not all).

**Limitations:**

The limitation of the work, would be the set of environments the algorithm was tested on. It would be interesting to see it tested on partially observable environments like Minecraft etc.

**Paper Formatting Concerns:**

No concerns.

**Quality:**

2

**Strengths And Weaknesses:**

Strengths:
The empirical validation demonstrates compelling evidence for the method's effectiveness across multiple dimensions. The authors conduct experiments on four different Procgen environments and show through statistical comparisons that their approach outperforms traditional dimensionality reduction methods like UMAP, PCA, and t-SNE on standard clustering metrics.

Weakness:
1) No Ground Truth: They never establish what "correct" semantic clustering should look like
2) The technical approach exhibits several design limitations that restrict its applicability. The number of clusters must be manually predetermined (they use K=8) with no adaptive mechanism to determine the optimal number for different environments or tasks.
3) A critical practical limitation is the continued reliance on manual interpretation despite claims of enhancing interpretability. The method requires humans to examine clusters and manually assign semantic meanings, with no automated mechanism for generating natural language descriptions or understanding what behaviors each cluster represents.

Improvements/suggestions:
1) Introduction needs a bit more detail of the method the authors are proposing. The contribution section states that the paper proposes a novel architecture that addresses limitations. It would be better to directly mention what this method is here and what is the novelty.
2) Missing related work:

* patil et. al: contrastive abstraction for RL
    * Uses contrastive learning to structure the space and uses hopfield network to obtain related clusters to a state.
* Jan Robine et.al: smaller world models for reinforcement learning
    * Use VQ-VAE to build world models. Related work for VQ-VAE and RL.

---

> ### Author Rebuttal · Authors · 2025-07-28
>
> # Rebuttal to Reviewer cNxc
>
> We sincerely thank Reviewer cNxc for their feedback and for highlighting our thorough empirical validation as well as the superior performance of our end-to-end semantic clustering approach compared to UMAP, PCA, and t‑SNE. Below, we address each concern.
>
> ## Weaknesses
>
> > 1. No Ground Truth: They never establish what "correct" semantic clustering should look like
>
> Since semantics are inherently subjective, there is no universal 'correct' clustering. As is standard in interpretability research, we rely on human evaluation (lines 203–219) rather than a fixed ground truth (Doshi‑Velez & Kim, *Toward a Rigorous Science of Interpretable ML*, 2017).  Our method achieves high inter-rater agreement on cluster semantics (Table 2), confirming consistent human interpretation of our clustering results.
>
> > 2. The technical approach exhibits several design limitations that restrict its applicability. The number of clusters must be manually predetermined (they use K=8) with no adaptive mechanism to determine the optimal number for different environments or tasks.
>
> Our clustering approach in this paper is consistent with the state-of-the-art works on this topic (e.g., Xie et al. (2016), Caron, et al. (2018)), which use fixed cluster numbers rather than adaptive mechanisms to determine the optimal number of clusters.  That being said, we agree that this is a limitation (lines 247–248) and will explore adaptive mechanisms (e.g., elbow method, silhouette-score optimization) in future work.
>
> >  3. A critical practical limitation is the continued reliance on manual interpretation despite claims of enhancing interpretability. The method requires humans to examine clusters and manually assign semantic meanings, with no automated mechanism for generating natural language descriptions or understanding what behaviors each cluster represents.
>
> We would like to note that the baseline works (Mnih et al. (2015) or Zahavy et al. (2016)) also use manual rather than automated description generation. Nevertheless, we agree that the reliance on manual generation of semantic descriptions is a limitation (lines 251–252), and will explore methods for automated generation of natural language descriptions (e.g., using GPT-4V) in future work.
>
> However, we disagree that the manual generation of description runs counter to our claim of enhancing interpretability. In a sense, these are orthogonal issues. Automated description generation could speed up the experiment design process, but would likely have little impact on the responses provided by the participants (who would not know whether the descriptions are generated manually or automatically), as automatically generated descriptions would still need to be manually verified by the researchers before being presented to the participants, given the possibility of hallucinations by VLMs (Liu et al. (2025))
>
> We also note that even with the continued reliance on manual description generation, our method drastically reduces annotation workload compared to baseline works. For example, we estimate that annotating a single environment via manual per‑state feature analysis and semantic grouping using the methods proposed by Mnih et al. (2015) or Zahavy et al. (2016) would require **hours**, whereas our annotators reviewed a handful of short video clips and generated cluster descriptions (like the ones in Table 1) in approximately **15  minutes**.
>
>
> ## Questions
>
> > 1. "cluster inputs based on their semantic similarity in the internal space": what is the internal space?
>
> The model's latent feature space.
>
> > 2. Why element wise addition of the feature vector? why not concatenate it?
>
> Unlike concatenation, element-wise addition does not change the original RL feature dimensionality. This enables integration without modifying downstream layers (lines 7, 78).
>
> > 3. If the goal is to analyse the DRL policy, then why make it conditional by adding to the policy term?
>
> Adding the discrete code to the policy head enables extension to downstream tasks (e.g., macro action selection in hierarchical learning) (see Fig. 1 caption, lines 140–142, 226–231).
>
> > What if the policy degrades?
>
> The policy does not degrade (see sections C.1,  D.1 and lines 153–154).
>
> > 4. The results show that sematic similar states are clusterd together. But what if the features which procgen is changing (background), is ignored by the feature generator already (before the clustering module)? Implying the added VQ-VAE is not necessary.
>
> Modified VQ‑VAE is necessary for both clustering and enhancing interpretability (lines 77–80), regardless of background.
>
> - Lines 118–122 and section B.2 show that the modified VQ‑VAE acts like online k‑means for clustering latent features.
> - Lines 126–138 and section C.2 show that when trained alongside the FDR loss, the modified VQ‑VAE loss sharpens cluster boundaries, directly boosting the interpretability of the DRL model.
>
> ## Improvements/suggestions
>
> Thank you for your suggested improvements regarding the introduction and related work! We will incorporate them into the camera-ready version.
>
> ## References
> - Xie, et al, Unsupervised Deep Embedded Clustering, ICML 2016
> - Caron, et al., Deep Clustering for Unsupervised Learning of Visual Features, ECCV 2018
> - Liu, et al., Reducing Hallucinations in Large Vision-Language Models via Latent Space Steering, ICLR 2025
> - Mnih, et al., Human-level control through deep reinforcement learning. Nature 2015.
> - Zahavy, et al., Graying the black box: Understanding DQN, ICML 2016

---

> > ### Comment · Reviewer_cNxc · 2025-08-06
> > **Response**
> >
> > Thank you for the detailed response and it has addressed most of my concerns. I have updated my score reflecting it.

---

> > > ### Author Response · Authors · 2025-08-06
> > >
> > > Thank you for taking the time to review our rebuttal and for updating your score. We’re pleased to hear that our response addressed your concerns and truly appreciate your thoughtful feedback.

---

### Decision · Program_Chairs · 2025-09-17

**Decision:**

Accept (poster)

**Comment:**

(a) This paper proposes a new semantic clustering module for DRL that integrates feature dimensionality reduction and online VQ-based clustering directly into the RL training loop. It is effectively doing online K-means (with VQ VAEs) within the policy loop. Unlike other interpretability works that used t-SNE or manually defined features, this approach is automatic and simple. In its current form, a discrete VQ is trained and element-wise added to the original feature vector, but this can be generalized to have entire discrete differentiable programs via LLMs. This line of work could be interesting from the perspective of learning interpretable policies end-to-end rather than some post or pre hoc feature based thinking.

(b) It is simple, end-to-end and can plug into existing DRL training pipelines. It also creates a trainable clustering mechanism that co-evolves with policy learning, which other methods fail to address. Experimental results are sound and validated through procgen environments, cluster quality metrics and human studies (more should be done). It also enables dynamic episode segmentation into semantic skills or options, which could have insights into hierarchical policy structure & learning.

(c) While the method reveals clear clusters, interpretation still requires human labeling of behaviors, and the paper lacks an automated mechanism for generating descriptions. The number of clusters must be manually set and no adaptive strategy is explored. Experiments are limited to four Procgen environments.

(d) To the best of my knowledge, no prior DRL interpretability work has embedded such clustering end-to-end. The most common baseline is t-SNE and it is interesting to carefully study and explore this line of work. Perhaps it has implications for end-to-end neuro symbolic learning. I am interested in seeing this work expand where the VQ is generalized to have more flexible structures (e.g. code gen via LLMs).

(e)
- Reviewers asked how correct "semantic" clustering is defined. But there is no universal gold standard. This is inherently emergent and subjective.
- Concerns were raised about fixed cluster # (K). Authors acknowledged this as a limitation
- Reviewers noted reliance on humans to assign semantics. Authors argued their method reduces workload drastically compared to prior state annotation and suggested to exploring automatic description generation. I think this can be independently studied and done on top of such methods with multi-modal LLMs.
- Multiple reviewers asked about loss design (FDR, VQ-VAE), element-wise addition, and policy conditioning. Authors provided detailed clarifications and theoretical proofs in appx, showing their modified VQ-VAE is equivalent to online k-means.

I believe all major questions were addressed during the rebuttal period.